J Physiol 604.1 (2026) pp 193–213

# Antenatal melatonin for cardiovascular deficits in fetal growth restriction

Charmaine R. Rock[1,2] , Tegan A. White[1,2], Beth R. Piscopo[1,2], Amy E. Sutherland[1,2], Yen Pham[1,2], Connor Karozis[1], Fiona L. Cousins[1,2], Emily J. Camm[1,2], Suzanne L. Miller[1,2] and Beth J. Allison[1,2]

[1] The Ritchie Centre, Hudson Institute of Medical Research, Clayton, VIC, Australia
[2] Department of Obstetrics and Gynaecology, Monash University, Clayton, VIC, Australia

Handling Editors: Laura Bennet & Janna Morrison

The peer review history is available in the Supporting Information section of this article (https://doi.org/10.1113/JP288750#support-information-section).

The Journal of Physiology

**Abstract figure legend** This study used an ovine model of fetal growth restriction (FGR) to investigate the potential cardioprotective effects of melatonin (MLT) in postnatal growth-restricted lambs. We demonstrated that, while peripheral endothelial function progressively declines between birth and 4 weeks in FGR lambs, FGR+MLT lambs exhibit impaired endothelial function at birth that improves by 4 weeks. However, MLT also increased oxidative stress and inflammation in the peripheral vasculature. Together, these results illustrate the multifaceted nature of melatonin's cardiovascular actions and emphasise the importance of further preclinical assessment to determine its therapeutic potential for the cardiovascular system (SUAL = single umbilical artery ligation).

S. L. Miller and B. J. Allison contributed equally to this work.

**Abstract** Fetal growth restriction (FGR) increases the risk of cardiovascular disease. FGR is linked to placental insufficiency and fetal hypoxemia, leading to oxidative stress and inflammation, which collectively influence the developmental programming of cardiovascular disease. This study assessed whether melatonin (MLT), a potent antioxidant and anti-inflammatory agent, could prevent cardiovascular deficits associated with FGR. Placental insufficiency was induced in ewes at 89 days of gestational age (dGA, term 148 dGA). Ewes were randomly allocated to control, FGR or FGR+MLT (I.v., 15 mg day$^{-1}$, from 95 dGA to birth) groups. Lambs were delivered preterm at 136 dGA and assessed as newborn (24 h) and 4-week-old lambs. Vascular function was determined in femoral arteries using *in vitro* wire myography and vascular morphology as assessed in carotid and femoral arteries. Newborn FGR lambs were ∼30% smaller than control lambs with an increased brain-to-body weight ratio, indicative of brain sparing. Femoral endothelial function declined between ∼24 h after birth and 4 weeks in FGR lambs. By contrast, femoral arteries from newborn FGR+MLT lambs displayed transient endothelial dysfunction that improved by 4 weeks. However, these arteries showed elevated levels of oxidative stress and inflammation. Despite improving endothelial function, melatonin also disrupted the brain-sparing response in FGR lambs. Furthermore, by 4 weeks of age, melatonin treatment led to heightened oxidative stress and inflammatory markers in the peripheral vasculature, suggesting a potential trade-off between vascular benefits and systemic maladaptation. These findings highlight the complexity of melatonin's effects on the cardiovascular system and underscore the need for careful evaluation of its long-term safety and efficacy before clinical translation.

(Received 16 February 2025; accepted after revision 27 October 2025; first published online 12 November 2025)
**Corresponding author** C. R. Rock: The Ritchie Centre, Hudson Institute of Medical Research, Clayton, VIC 3168, Australia. Email: charmaine.rock@monash.edu

**Key points**

- Fetal growth restriction (FGR) significantly increases the lifelong risk of cardiovascular disease, and there are currently no targeted treatments to mitigate these risks.
- This study follows growth-restricted lambs from birth to 4 weeks of age (comparable to a 1-year-old human in terms of cardiovascular function) to characterise how FGR affects vascular development over time.
- FGR lambs exhibited progressive endothelial dysfunction in the femoral artery, but antenatal melatonin treatment restored endothelial function long-term despite the presence of vascular oxidative stress and inflammation.
- The brain-sparing response is a key adaptive mechanism for fetal survival, yet melatonin appears to dampen this response, highlighting the need for further investigation into its broader physiological effects.

**Charmaine R. Rock** is an early career researcher who completed her PhD in the Perinatal Cardiovascular Physiology and Neurodevelopment and Neuroprotection research groups at the Hudson Institute of Medical Research and Monash University. Her research explores how pregnancy complications, such as fetal growth restriction and prematurity, affect cardiovascular development and adaptation. She is currently expanding her work at Vanderbilt University Medical Center, investigating the mechanisms regulating ductus arteriosus closure in the perinatal period and factors that may impair this process.

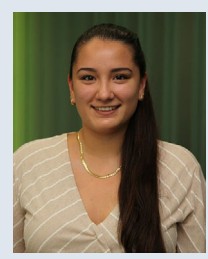

## Introduction

Fetal growth restriction (FGR) is a severe pregnancy complication where the fetus fails to thrive and does not achieve its genetic growth potential (Gordijn et al., 2016). The aetiology of FGR may be fetal, maternal or placental in origin, with the primary cause being placental insufficiency (Malhotra et al., 2019). Placental insufficiency impairs oxygen and nutrient delivery to the fetus, creating a hypoxemic intrauterine environment. In turn, fetal hypoxemia induces an adaptive cardiovascular response to redistribute cardiac output to increase blood flow towards essential organs at the expense of non-essential organs and tissues (Allison et al., 2016). This cardiovascular adaptation is termed brain sparing and is mediated by chronic vasoconstriction of peripheral vascular beds during fetal gestation (Giussani, 2016). Although brain sparing is a crucial adaptation for fetal survival, clinical and preclinical studies indicate that chronic exposure to hypoxemia *in utero* and subsequent brain-sparing results in altered heart and vascular development (Rock et al., 2023; Sehgal et al., 2018, 2019, 2023). Impaired development of the cardiovascular morphology *in utero* leads to impaired cardiac function, which heightens the lifelong risk of cardiovascular disease (Herrera et al., 2025). Subtle and subclinical cardiovascular risk factors such as increased cardiac globularity and vascular intima-media thickness are detectable as early as the first days of life in infants who experienced FGR.

To date, the mechanisms programming for cardiovascular dysfunction in those born growth-restricted have remained poorly described, probably as a result of the multiple causative factors underlying FGR (Rock et al., 2021). One contributor is chronic vasoconstriction of the peripheral vasculature, which leads to a cascade of processes increasing systemic vascular resistance, resulting in a stiffer and less compliant cardiovascular system when sustained long-term. Strong evidence also suggests a role for inflammation and oxidative stress in the pathogenesis of cardiovascular dysfunction in FGR. Oxidative stress not only causes cellular DNA damage and degradation, but also impairs blood vessel function by modulating nitric oxide (NO) (Förstermann, 2010). In blood vessels, NO is released from the endothelium and regulates vascular tone; however, when excess superoxide is present, as occurs with oxidative stress, NO rapidly binds to superoxide and produces the more potent free radical, peroxynitrite. This causes a local reduction in NO bioavailability in the vasculature, increasing blood vessel tone and contributing to the chronic vasoconstriction initiated by the brain-sparing response (Giussani, 2016). The long-term implications of chronically reduced NO bioavailability are unknown. Additionally, peroxynitrite further amplifies oxidative stress in a feedback loop that overproduces free radicals and induces a fetoplacental inflammatory response via upregulation of pro-inflammatory cytokines. Although brain-sparing, oxidative stress and inflammation each have direct and indirect actions on the vasculature; combined, they probably have an additive effect on adverse vascular structure and function. Thus, any potential therapy to combat cardiovascular deficits in FGR should incorporate antioxidant and anti-inflammatory properties.

Melatonin is a naturally occurring hormone with potent antioxidant and anti-inflammatory properties (Hardeland et al., 2011). The therapeutic benefits of melatonin are of interest in complications of pregnancy, particularly for its neuroprotective benefits, with a good safety profile for pregnancy and the ability to cross the placenta (Okatani et al., 1998). Melatonin acts through receptor-mediated and receptor independent pathways. Its receptor-mediated actions are driven by G-protein-coupled melatonin receptors ($MT_1$ and $MT_2$) in many target organs, which, in blood vessels, are expressed in both the endothelium and smooth muscle (Liu et al., 2016). Activation of these receptors modulates redox homeostasis by upregulating enzymes such as superoxide dismutase and inhibiting pro-oxidant enzymes, including NADPH oxidase. Although melatonin is often described as a direct free radical scavenger, the *in vivo* evidence for this action is limited (Liu et al., 2016; Monteiro et al., 2024; Paulis et al., 2012) and current data suggest that its vascular protective actions are more plausibly explained by receptor-mediated enhancement of endogenous antioxidant defenses rather than direct interactions with reactive oxygen species (ROS). Our recent systematic review demonstrated that melatonin has been the most frequently investigated antioxidant therapy for cardiovascular dysfunction in preclinical studies of FGR, demonstrating a strong capacity to reduce oxidative stress and improve endothelial dysfunction in peripheral vascular beds (Rock et al., 2024). For example, in chick embryos incubated in a hypoxic environment (14% $O_2$), antenatal melatonin administration reduced the oxidative stress markers 3-nitrotyrosine and 4-hydroxynonenal, and also increased antioxidant enzymes, including glutathione peroxidase, in the fetal heart (Itani et al., 2016). Antenatal melatonin treatment has also been shown to improve vascular endothelial function by increasing NO bioavailability in isolated femoral arteries from chicken embryos (Itani et al., 2016) and coronary arteries from newborn lambs (Tare et al., 2014). Thakor et al. (2015) demonstrated that melatonin administration diminished peripheral vasoconstriction during acute fetal hypoxemia, which was facilitated by an increase in NO bioavailability in sheep (Doolen et al., 1998). Thus, by increasing NO bioavailability and allowing NO to act as a vasodilator, melatonin may alleviate peripheral vaso-

constriction. However, on balance, it remains unknown whether a dampening of the brain-sparing response by melatonin to improve NO bioavailability in the FGR fetus is beneficial or harmful, and the longer-term effects on the cardiovascular system remain unexplored.

Subclinical vascular dysfunction is measurable in FGR infants soon after birth (Sehgal et al., 2018, 2019, 2023). Preclinical studies in growth-restricted fetal and neonatal sheep demonstrate a differential profile of altered cardiovascular development from fetal to postnatal FGR life (Rock et al., 2023). The present study aimed to determine whether antenatal maternal melatonin treatment normalises vascular development in FGR offspring. To achieve this aim, femoral vascular function and morphology in carotid and femoral arteries collected from newborn and 4-week-old control lambs, FGR lambs and FGR lambs exposed to maternal antenatal melatonin were assessed. Additionally, potential causative pathways contributing to the development of cardiovascular dysfunction were investigated. It was hypothesised that antenatal treatment with melatonin would ameliorate cardiovascular deficits in growth-restricted lambs predominantly mediated via its antioxidant and anti-inflammatory actions.

## Methods

### Ethical approval

Experimental protocols were approved by the Monash Medical Centre Animal Ethics Committee (MMCA 2016/19 and 2019/02) in accordance with guidelines established by the National Health and Medical Research Council of Australia and the 'Animal Research: Reporting of In vivo Experiments Guidelines' (Percie Du Sert et al., 2020).

### Animal care and surgical preparation

Singleton-bearing mixed-breed pregnant ewes were delivered to Monash Health Translation Precinct Animal Facility and housed in individual pens in a room maintained at 22°C and 44–55% relative humidity with a 12:12 h light/dark photocycle. All ewes were fed ∼1 kg of lucerne chaff daily and had free access to water. Food intake and general wellbeing of the ewe were monitored daily and any abnormal behaviour or symptoms were treated accordingly. At the time of surgery, ewes were randomly allocated to control ($n_{newborn} = 8$, $n_{4-week} = 11$), FGR ($n_{newborn} = 9$, $n_{4-week} = 9$) or FGR+melatonin (FGR+MLT; $n_{newborn} = 7$, $n_{4-week} = 8$) groups. Sterile surgery was performed at 89 days of gestational age (dGA, term 148 dGA) to induce early onset FGR via single umbilical artery ligation (SUAL) as previously described

(Rock et al., 2023; Sutherland et al., 2024). Anaesthesia was induced with an injection of sodium thiopentone via the jugular vein (20 mL of Pentothal: Jurox, Rutherford, NSW, Australia) and maintained after intubation with a cuffed endotracheal tube (inner diameter 8.0 mm, outer diameter 10.9 mm) (Portex; ICU Medical, San Clemente, CA, USA) with isoflurane (1.5–2.5% in oxygen). A laparotomy was performed to access the uterus, and SUAL was performed to induce FGR (FGR and FGR+MLT) or the umbilical cord was manipulated but not ligated (control). A maternal jugular vein catheter was inserted for antibiotic and melatonin administration. Antibiotics (1 g of Ampicillin; Austrapen, Lennon Healthcare, Ives, NSW, Australia; 5 mL of Engemycin; Coopers, Bendigo East, Vic, Australia) were administered to the ewe via the jugular vein immediately following the induction of anaesthesia and were maintained for 3 days post-surgery. Additionally, all ewes received paracetamol for analgesia at the cessation of surgery (rectal, 500 mg) followed by repeated doses for 3 days (oral, 1 g) (Panadol, Waterford, Ireland). After surgery, ewes were returned to their pens and their recovery monitored.

At 95 dGA, a continuous I.V. infusion of melatonin [15 mg day$^{-1}$ (Sigma-Aldrich, Castle Hill, NSW, Australia) in absolute ethanol (1%) and 0.9% saline] commenced to the ewe in FGR+MLT animals until birth. This dose was derived from a previous clinical investigation (Hobson et al., 2018) and aligns with the current PROTECTMe clinical trial, in which antenatal melatonin is administered to early-onset FGR pregnancies for neuroprotection (Palmer et al., 2019). At 134 dGA. All ewes were induced for preterm birth using I.M. injections of mifepristone (50 mg of Linepharma; MS Health, Richmond, Vic, Australia) and betamethasone (11.4 mg of celestone chronodose; Organon Pharma, Macquarie Park, NSW, Australia, repeated 24 h later). Labouring ewes were monitored remotely via video, and lambs were born at 136 dGA with researchers present to weigh lambs and monitor their wellbeing.

### Lamb monitoring

Lambs were checked regularly in the hours after birth and then daily until 4 weeks of age. Non-invasive blood pressure measurements were collected on the first day of life and before postmortem with an appropriately sized blood pressure cuff (UI-0515; Technicuff, Leesburg, FL, USA) connected to a multiparameter vital signs monitor (Surgivet, Lasne, Belgium) as previously described (Rock et al., 2023).

### Postmortem and tissue collection

Lambs were weighed the day after birth and at 4 weeks of age, and plasma samples were obtained before death

(sodium pentobarbitone, 10 mL of i.v.; Virbac, Milperra, NSW, Australia). Organ weights were recorded. Whole hearts were fixed in 10% neutral-buffered formalin for assessment of cardiac dimensions. Segments of the first-order femoral and carotid arteries ($\sim$2 cm) were dissected and fixed in 10% neutral-buffered formalin for histological and immunohistochemical analysis. For *in vitro* wire myography, the right hind leg was removed and immediately placed in chilled working Krebs solution, consisting of (in mmol $L^{-1}$) 118.845 NaCl, 4.69 KCl, 1.156 $MgSO_4$, 1.175 $KH_2PO_4$, 25.2 $NaHCO_3$, 12 D-glucose, 0.03 EDTA and 2.5 $CaCl_2$.

### *In vitro* wire myography

*In vitro* wire myography was conducted on third-order femoral arteries as previously described (Rock et al., 2023). Briefly, third-order femoral arteries were mounted on a wire myograph (Multi Wire Myograph System 610M; DMT, Hinnerup, Denmark) containing Krebs solution, bubbled with carbogen and normalised (90% of 5.3 kPA; DMT normalisation module) before endothelial vascular reactivity was confirmed. Vasodilatation response curves to sodium nitroprusside (SNP) ($10^{-10-4}$ mol $L_{-1}$; Sigma, St Louis, MO, USA) and ACh ($10^{-10-5}$ mol $L^{-1}$; Sigma) were determined after submaximal pre-contraction with serotonin (5 Ht). Response curves were repeated in the presence of $N^{\omega}$-nitro-L-arginine methyl ester (L-NAME) ($10^{-5}$ mol $L^{-1}$; Sigma) and indomethacin ($10^{-6}$ mol $L^{-1}$; Sigma). Concentration-response curves were graphed as percentage tension present in the vascular bed and analysed using an agonist-response line of best fit. The maximal relaxant response ($R_{max}$) was calculated from the line of best fit and expressed as the percentage difference between active tone before the addition of the vasodilator and after the administration of the highest dose of the vasodilator ($10^{-5}$ mol $L^{-1}$ ACh or $10^{-4}$ mol $L^{-1}$ SNP). The $EC_{50}$, which is the concentration of agonist that provokes a response halfway between the baseline and maximum response of the vasodilatation curve, was calculated from the line of best fit and vascular sensitivity was expressed as $pD_2$ ($-\log^{EC50}$) (Rock et al., 2023).

### Circulating melatonin and oxidative stress

Melatonin was extracted from plasma using an extraction kit (MLT-EXSET; NovoLytiX, Bättwil,, Switzerland) and the concentration was determined using enzyme-linked immunosorbent assay (ELISA) (MLTN-96-U; NovoLytiX). Plasma malondialdehyde (MDA) concentration, a marker of lipid peroxidation, was measured using a TBARS assay kit (10009055; Cayman Chemical, Ann Arbor, MI, USA). All extractions and assays were performed as per the manufacturer's instructions. The intra- and inter-assay coefficients of variation of the melatonin ELISA were 7.5% and 11.1%, respectively. The intra- and inter-assay coefficients of variation of the MDA assay were 7.4% and 23.7%, respectively.

### Cardiac dimensions

To assess cardiac dimensions, the whole formalin-fixed heart with the pulmonary trunk facing upward, was photographed using an iPad (Apple, Cupertino, CA, USA) on a stereological grid (5 $\times$ 5 mm) to allow the image size to be determined. The image was then exported into ImagePro Premier 10 (Media Cybernetics, Rockville, MD, USA) to quantify cardiac length (mm) and width (mm) relative to the grid. The shape of the heart was outlined using the polygon tool, and the circularity measurement in ImagePro Premier 10 [circularity = $(4 \cdot area)/(\pi \cdot MaxFeret2)$] was used to determine cardiac globularity. The heart was then transversely dissected and cut into 1 cm thick sections from the apex through to the top of the heart. These sections were then laid cut-side up and imaged. The thickness (mm) of the right ventricle wall, left ventricle wall, and septal wall was measured on ImagePro Premier 10, using an average of three measurements from each wall. The average wall thickness was corrected for the total diameter of the heart per section. All cardiac measurements were corrected for heart weight.

### Histological and immunohistochemical analysis of the vasculature

Formalin-fixed and paraffin-embedded carotid and femoral arteries were sectioned (4 μm) to assess the composition of the extracellular matrix proteins in the tunica media. Sections were stained with picrosirius red to identify collagen, as well as Hart's resorcin-fuchsin stain for elastin (Joyce et al., 2003) within the tunica media. Immunohistochemistry was used to evaluate vascular smooth muscle integrity [$\alpha$-smooth muscle actin ($\alpha$SMA)], angiogenesis [vascular endothelial growth factor (VEGF)], inflammation [C-reactive protein (CRP)], oxidative stress [8-hydroxy-2'-deoxyguanosine (8-OHdG)] and components of the catecholamine pathway ($\alpha_{1A}$-adrenoreceptors, $\alpha_{1A}$-AR and tyrosine hydroxylase) in the tunica media. In brief, sections were dewaxed and rehydrated in serial alcohols (Castillo-Melendez et al., 2017). Antigen retrieval, serum blocking, and primary and secondary antibody incubations are outlined in Table 1. Biotinylated secondary antibody was detected with streptavidin horseradish peroxidase (StrepHRP; dilutin 1:200, Amersham Bioscience, Little Chalfont, UK) and developed with

**Table 1. Immunohistochemical methodology**

| | Antigen retrieval | Peroxidase block | Protein block | Primary antibody | Secondary antibody | Method of analysis |
|---|---|---|---|---|---|---|
| **α-Smooth muscle actin (αSMA)** | DAKO Target Retrieval (Low pH) Solution (#S1699) 30 min at 98°C (DAKO PT Link) | DAKO Peroxidase Blocking Solution (#S2023) 10 min at RT | DAKO serum-free protein block 30 min at RT | Monoclonal Mouse anti-SMA (#M0851) in DAKO antibody diluent (#S0809) 1:200 dilution 60 min at RT | DAKO Envision+ System HRP Labelled Polymer anti-mouse (#K4001) 30 min at RT | % area positive staining |
| **C-reactive protein (CRP)** | 0.01 M Sodium citrate buffer (pH 6) 1 × 3 min on high and 2 × 3 min on low (microwave) | 3% $H_2O_2$ in methanol 10 min at RT | DAKO serum-free protein block 60 min at RT | Polyclonal rabbit CRP (#MBS2111991) in DAKO antibody diluent 1:200 dilution overnight at 4°C | Goat anti-rabbit 1:200 dilution 60 min at RT | % area positive staining |
| **8-hydroxy-2′-deoxyguanosine (8-OHdG)** | 0.01 M Citric acid buffer (pH 6) 3 × 5 min on high (microwave) | 3% $H_2O_2$ in 50% methanol/$dH_2O$ 15 min at RT | 5% NGS/3% BSA/0.1% Tx in PBS 45 min at RT | Monoclonal mouse anti-8-OHdG (#MOG-100P) in 2% NGS/1% BSA/0.1% Tx in PBS 1:200 dilution overnight at 4°C | Goat anti-mouse 1:200 dilution 60 min at RT | Number of positive cells per $mm^2$ |
| **Vascular endothelial growth factor (VEGF)** | 1 mM EDTA, 0.05% Tween20 (pH 9) 30 min at 98°C (DAKO PT Link) | DAKO Peroxidase Blocking Solution (#S2023) 10 min at RT | DAKO serum-free protein block 30 min at RT | Monoclonal mouse VEGF (#NB100-648) in DAKO antibody diluent 1:100 dilution overnight at 4°C | DAKO Envision+ System HRP Labelled Polymer anti-mouse 30 min at RT | % area positive staining |
| **α1A-adrenergic receptor (α1A-AR)** | 0.01 M Sodium citrate buffer (pH 6) 3 × 5 min on high (microwave) | 3% $H_2O_2$ in $dH_2O$ 10 min at RT | 5% NGS/3% BSA in PBS 45 min at RT | Polyclonal rabbit α1A-AR (#OASG00518) in 3% NGS in PBS 1:300 dilution overnight at 4°C | Goat anti-rabbit 1:200 dilution 45 min at RT | % area positive staining |
| **Tyrosine hydroxylase (TH)** | 0.01 M Sodium citrate buffer (pH 6) 9 min on high and 7 min on low (microwave) | 0.3% $H_2O_2$ in 50% methanol/$dH_2O$ 20 min at RT | CAS-Block™☐ (neat) 30 min at RT | Monoclonal mouse TH (#MAB318) in PBS 1:500 dilution overnight at 4°C | Goat anti-mouse 1:200 dilution 60 min at RT | % area positive staining |

Abbreviations: BSA, bovine serum albumin; NGS, normal goat serum; PBS, phosphate-buffered saline; RT, room temperature; Tx, triton X-100.

diaminobenzidine (MP Biomedicals, Irvine, CA, USA) and hydrogen peroxide (ChemSupply, Gillman, SA, Australia). All slides were coded, with qualitative assessment performed by a single examiner (CRR) who was blinded to the experimental group.

To assess intima-media thickness, slides stained for elastin were scanned at 40× using a VS120 slide scanner (Olympus, Tokyo, Japan) and the Monash Histology Platform (Clayton, VIC, Australia) and uploaded onto HALO imaging software (Indica Labs, Albuquerque, NM, USA). The 'DenseNet V2' classifier in HALO was trained on multiple example regions to determine the area of the lumen (a), intima-media (b) and total tissue area. Using these measurements, intima-media thickness ($x$) was determined using: $x = \sqrt{\frac{a+b}{\pi}} - \sqrt{\frac{b}{\pi}}$. This calculation relies on the cross-section approximating a circle, and therefore any sections that were flattened during tissue processing were excluded from analysis. Intima-media thickness was corrected for the total tissue area. Two cross-sections were assessed for each animal and the intima-media thickness values were averaged.

Slides stained for $\alpha$SMA were scanned at 40× using an VS120 slide scanner (Olympus) and the Monash Histology Platform and uploaded to QuPath. The intima-media and adventitia were manually annotated and corrected for the total tissue area to determine the percentage area of each layer. The percentage area of positive $\alpha$SMA staining in the intima-media was determined using a thresholder in QuPath (https://qupath.github.io). For all other histological and immunohistochemical analyses, images were taken using a BX41 microscope with a DP25 Camera (Olympus). Ten fields of view within the tunica media were imaged with a 100× objective. 8-OHdG positive cells were manually counted in QuPath, and all other stains were analysed using a thresholder in QuPath, where the percentage area of the positive staining was determined per field of view. Results from CRP, 8-OHdG, VEGF, $\alpha_{1A}$-AR and tyrosine hydroxylase were corrected for total vessel area.

### Statistical analysis

Statistical analyses were conducted using Prism, version 10 (GraphPad Software, Boston, MA, USA). Data were assessed for normality using Shapiro–Wilk test and are expressed as the mean $\pm$ SD unless otherwise stated. Dose-response curves from *in vitro* wire myography were fitted with Sigmoidal curves and the relative contributions of NO, prostanoids, and endothelium-derived hyperpolarisation (EDH) were determined through area under the curve (AUC) analysis. $R_{max}$ and $EC_{50}$ are displayed as the mean $\pm$ confidence interval. In the first instance, we assessed gestational age changes separately for each parameter in the control, FGR and FGR+MLT groups using an unpaired $t$ test (or a Mann–Whitney $U$ test for data not normally distributed). This was followed by a one-way analysis of variance (ANOVA) with a *post hoc* Tukey test, where applicable on the same variables to determine the effects of the independent variables of treatment. When data were not normally distributed, a Kruskal–Wallis test was used with a *post hoc* Dunn's test. $P < 0.05$ was considered statistically significant.

## Results

### Lamb biometrics

Lamb characteristics are described in Table 2. At birth, FGR lambs weighed less than control lambs in both the newborn ($P = 0.0081$) and 4-week-old groups ($P = 0.0006$). The birth weight of newborn FGR+MLT lambs was not significantly different from either the newborn control ($P = 0.102$) or newborn FGR lambs ($P = 0.575$). However, the birth weight of FGR+MLT lambs in the 4-week-old cohort was significantly lower than that of controls in the same cohort ($P = 0.039$). Treatment with melatonin significantly delayed the birth time of the newborn FGR lambs by 7 h (control: 07.00 h $\pm$ 2.2 h *vs.* FGR: 09.00 h $\pm$ 5 h *vs.* FGR+MLT: 14.00 h $\pm$ 3.2 h, $P_{control}$ *vs.* FGR+MLT = 0.01). At post-mortem, the day after birth, both newborn FGR and FGR+MLT lambs remained significantly smaller than control lambs ($P = 0.004$ and $P = 0.033$, respectively). Evidence of asymmetric growth with increased brain:body weight ratio was present in FGR newborns ($P = 0.011$) but not FGR+MLT newborn lambs ($P = 0.223$). At 4 weeks of age, the body weight of FGR lambs remained reduced ($P = 0.014$) with an increased brain:body weight ratio compared to control lambs ($P = 0.018$). By contrast, the body weight of melatonin-treated lambs was not different at 4 weeks compared to either the control group or FGR group ($P = 0.082$). Furthermore, the brain:body weight ratio of 4-week-old FGR+MLT lambs was significantly lower than that of 4-week-old FGR lambs ($P = 0.036$). There was no evidence of 'heart-sparing' as heart:body weight ratio was similar across all groups at both ages.

Non-invasive blood pressure was measured in all lambs prior to postmortem, with no differences between the groups observed at either age (Table 2). However, the trajectory of blood pressure changes between the day after birth and 4 weeks of age in control, FGR and FGR+MLT lambs was also assessed. Mean blood pressure increased by 28 and 25 mmHg between birth and 4 weeks of age in control and FGR+MLT lambs ($P = 0.0017$ and $P = 0.0014$, respectively); however, no significant change was observed over the same period in FGR lambs ($P = 0.152$).

Melatonin concentrations were assessed in plasma collected from lambs immediately prior to death.

**Table 2. Lamb characteristics**

| | Newborn | | | | 4-week-old | | | |
|---|---|---|---|---|---|---|---|---|
| | Control | FGR | FGR+MLT | P value | Control | FGR | FGR+MLT | P value |
| **Number of animals** | 8 | 9 | 7 | – | 11 | 9 | 8 | – |
| **Sex** | 5 females, 3 males | 4 females, 5 males | 6 females, 1 male | – | 4 females, 7 males | 4 females, 5 males | 3 females, 5 males | – |
| **Birth weight (kg)** | 4.6 ± 0.9 | 3.1 ± 0.9* | 3.6 ± 0.9 | **P = 0.0100** | 4.4 ± 0.4 | 3.2 ± 0.7* | 3.7 ± 0.8* | **P = 0.0007** |
| **Mean blood pressure (mmHg)** | 56.9 ± 5.4 | 63.7 ± 12.0 | 52.1 ± 9.1 | P = 0.1129 | 77.7 ± 13.8 | 75.6 ± 16.9 | 76.3 ± 9.7 | P = 0.9611 |
| **Systolic blood pressure (mmHg)** | 77.7 ± 7.9 | 83.1 ± 12.4 | 74.2 ± 11.7 | P = 0.3693 | 106.1 ± 21.7 | 101.8 ± 18.5 | 102.5 ± 13.7 | P = 0.8940 |
| **Diastolic blood pressure (mmHg)** | 46.6 ± 6.4 | 54.1 ± 12.6 | 40.1 ± 7.9 | P = 0.0590 | 63.4 ± 10.6 | 63.2 ± 17.4 | 63.4 ± 8.5 | P = 0.9995 |
| **Postmortem** | | | | | | | | |
| **Body weight (kg)** | 5.0 ± 1.0 | 3.3 ± 1.0* | 3.6 ± 1.0* | **P = 0.0038** | 13.3 ± 1.3 | 11.1 ± 1.5* | 12.9 ± 2.2 | **P = 0.0143** |
| **Brain:body weight (g kg$^{-1}$)** | 11.0 ± 2.4 | 17.5 ± 5.7* | 14.85 ± 2.7 | **P = 0.0145** | 5.56 ± 0.5 | 6.5 ± 0.7* | 5.6 ± 0.9† | **P = 0.0122** |
| **Heart:body weight (g kg$^{-1}$)** | 8.8 ± 1.1 | 8.8 ± 0.9 | 9.3 ± 1.4 | P = 0.5475 | 6.9 ± 1.2 | 7.1 ± 1.3 | 7.03 ± 1.2 | P = 0.9263 |
| **Circulating factors** | | | | | | | | |
| **Melatonin (pg mL$^{-1}$)** | 17.6 ± 7.6 | 12.2 ± 11.5 | 256.5 ± 144.4*† | **P < 0.0001** | 4.6 ± 3.8 | 6.2 ± 2.6 | 7.4 ± 7.7 | P = 0.5237 |
| **Malondialdehyde (μM)** | 10.2 ± 2.0 | 14.5 ± 2.1* | 13.2 ± 2.9 | **P = 0.0033** | 8.4 ± 3.4 | 7.8 ± 2.4 | 7.5 ± 6.2 | P = 0.7529 |

Mean ± SD values are presented for each group. Analysed with one-way ANOVA with a *post hoc* Tukey test, when applicable. Bold was used to indicate a *p*-value < 0.05.
* $P < 0.05$, vs. Control. † $P < 0.05$, vs. FGR.

Newborn FGR+MLT lambs had higher circulating plasma melatonin compared to control and FGR lambs ($P < 0.0001$), which was normalised by 4 weeks of age. MDA was used as a marker of systemic oxidative stress, and newborn FGR lambs had elevated plasma MDA concentration compared to control lambs ($P = 0.003$), whereas newborn FGR+MLT demonstrated a trend towards increased plasma MDA compared to newborn control lambs ($P = 0.051$).

### Isolated femoral artery function

The function of third-order femoral arteries was evaluated via *in vitro* wire myography to determine endothelium independent and -dependent vasodilatory capacity. Endothelium independent vasodilatation was assessed using the NO donor, SNP (Fig. 1*A* and *B*). The maximal relaxation elicited to the highest dose of SNP was not different between groups at either age (Fig. 1*C*). When assessing vascular responsivity across development, control lambs demonstrate improved maximal vasodilatation in response to SNP with development ($P = 0.0255$), a difference not observed in growth-restricted lambs ($P = 0.169$). By contrast,

maximal endothelium independent vasodilatation in FGR+MLT lambs significantly decreased across the ages assessed ($P = 0.036$). FGR+MLT femoral arteries displayed a significant reduction in sensitivity to SNP compared to control and FGR lambs in the newborn ($P = 0.0002$) (Fig. 1*D*) and at 4 weeks of age ($P = 0.0007$). There was no difference in the sensitivity to SNP between FGR and control femoral arteries.

Femoral resistance arteries were exposed to increasing concentrations of ACh to determine endothelium-dependent vasodilatory capacity (Fig. 2*A* and *B*). Control and FGR lambs showed no developmental changes in endothelium-induced vasorelaxation, FGR+MLT lambs demonstrated an increased capacity to elicit vasodilatation through the endothelium across the times assessed ($P = 0.045$). Maximal relaxation to ACh was not different between newborn control and FGR animals; however, by 4 weeks of age, FGR lambs displayed a 31% reduction in maximal relaxation compared to control animals ($P = 0.017$) (Fig. 2*C*). By contrast, newborn FGR+MLT lambs exhibited a significant reduction in maximal relaxation to ACh ($P = 0.007$), which was no longer evident by 4 weeks of age. Endothelial sensitivity to ACh in femoral arteries was not different between groups at either age (Fig. 2*D*).

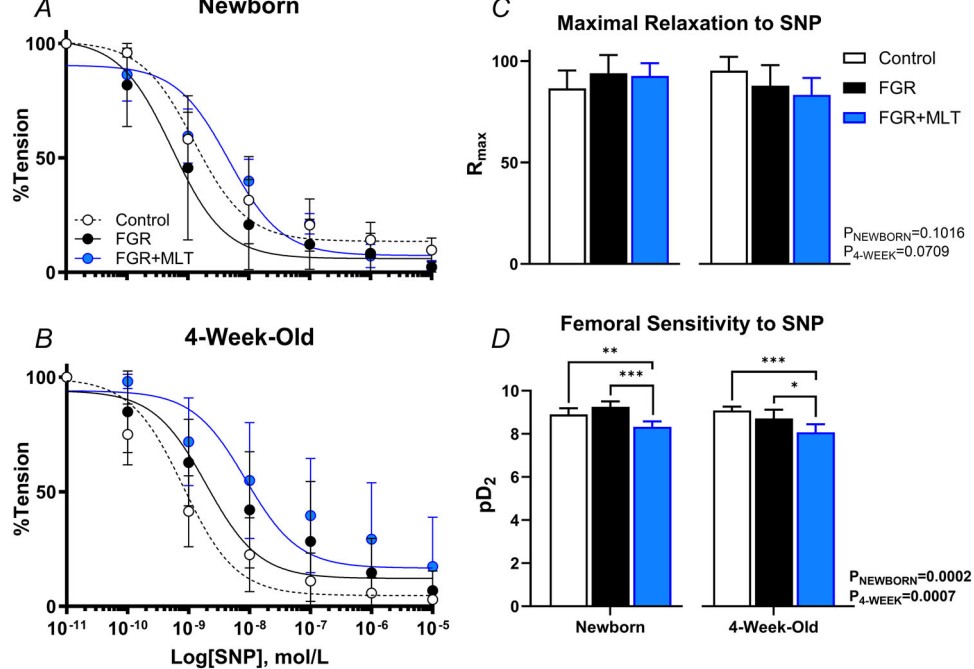

**Figure 1. Endothelium independent vasodilatation in isolated femoral arteries.**
Concentration–response curves to sodium nitroprusside (SNP) in third-order femoral resistance arteries from newborn (*A*) and 4-week-old (*B*) lambs are presented as the mean ± SD and fitted with a sigmoidal curve with data representing vasodilatation from submaximal contraction. Maximal relaxation ($R_{max}$) (*C*) and sensitivity (pD$_2$) (*D*) to SNP are displayed as the mean ± confidence interval and analysed using one-way ANOVA within age groups, with a *post hoc* Tukey test when applicable. Data presented in (*C*) and (*D*) are derived from a fitted sigmoidal curve, and therefore, individual data points are not available for display. Newborn: control (*n* = 5), FGR (*n* = 5), FGR+MLT (*n* = 7); 4-week-old: control (*n* = 5), FGR (*n* = 6) and FGR+MLT (*n* = 8). *$P < 0.05$, **$P < 0.01$, ***$P < 0.001$.

The function of the major endothelium-dependent vasodilatory pathways in the femoral artery was calculated through a series of pharmacological blockers (Fig. 3). Between the day after birth and 4 weeks of age, control lambs demonstrated a 6.6-fold increase in NO contribution to vasodilatation ($P = 0.0002$). This developmental change was not observed in FGR or FGR+MLT lambs ($P = 0.074$ and $P = 0.078$, respectively). However, NO contribution to vasodilatation in newborn FGR+MLT lambs was greater than that of control or FGR lambs ($P = 0.008$) (Fig. 3*A*). Newborn FGR+MLT lambs displayed a significant reduction in EDH contribution compared to FGR lambs ($P = 0.031$) (Fig. 3*C*). By 4 weeks of age, the improved NO handling observed in newborn FGR+MLT lambs had reversed, whereby there was a significant decrease in NO contribution compared to control lambs ($P = 0.005$). There were no differences in prostanoid (Fig. 3*B*) contribution to endothelium-dependent vasodilatation between groups at either age.

## Cardiovascular morphology

Cardiac morphology, along with carotid and femoral artery morphology, was assessed (Table 2). Assessment of cardiac morphology showed no significant differences between the groups at either age. In control animals, intima-media thickness decreased with age in both the carotid ($P = 0.043$) and femoral ($P = 0.018$) arteries. By contrast, no difference with ageing occurred in FGR and FGR+MLT lambs. There were no differences between groups for carotid and femoral artery dimensions (intima-media area, adventitia area or intima-media thickness).

Vascular smooth muscle abundance ($\alpha$SMA) (Table 3) in the carotid and femoral arteries was susceptible to alterations in FGR lambs. In control lambs, carotid smooth muscle abundance did not change with age, but decreased with age in the femoral artery ($P = 0.025$). FGR lambs showed a significant increase in the density of vascular smooth muscle in the carotid artery ($P = 0.033$), with no change in the femoral with

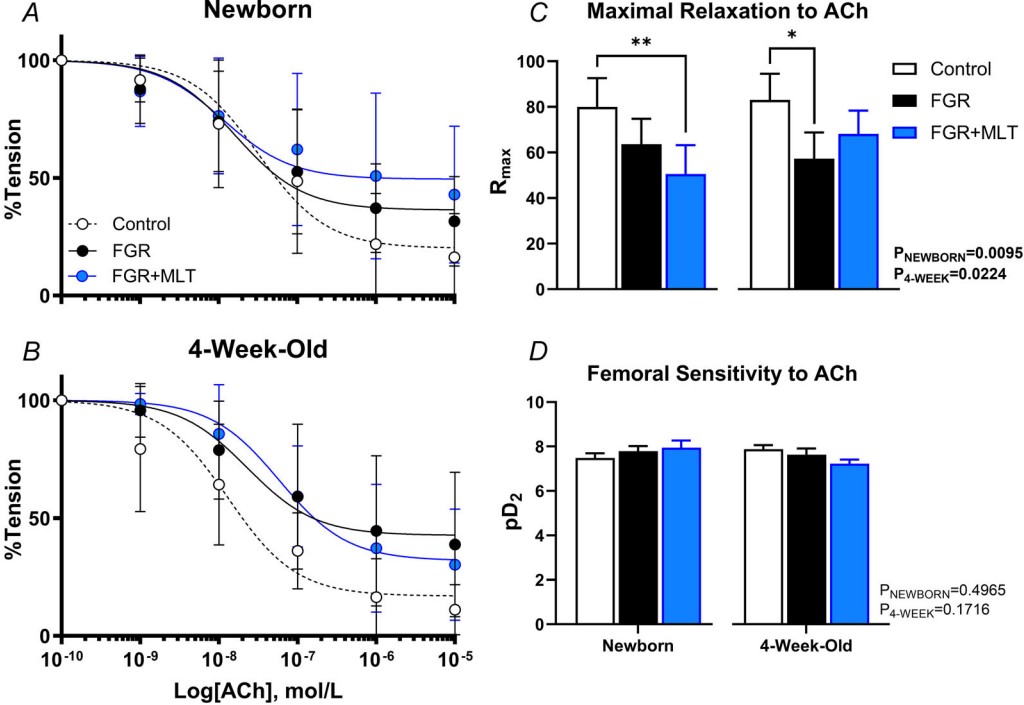

**Figure 2. Endothelium-dependent vasodilatation in isolated femoral arteries**
Concentration-response curves to acetylcholine (ACh) in third-order femoral resistance arteries from newborn (*A*) and 4-week-old (*B*) lambs are presented as the mean ± SD and fitted with a sigmoidal curve with data representing vasodilatation from submaximal contraction. Maximal relaxation ($R_{max}$) (*C*) and sensitivity (pD$_2$) (*D*) to ACh are displayed as the mean ± confidence interval and are analysed using one-way ANOVA within age groups, with a *post hoc* Tukey test when applicable. Data presented in (*C*) and (*D*) are derived from a fitted sigmoidal curve, and therefore, individual data points are not available for display. Newborn: control (*n* = 6), FGR (*n* = 5), FGR+MLT (*n* = 7); 4-week-old: control (*n* = 5), FGR (*n* = 8) and FGR+MLT (*n* = 8). *$P < 0.05$, **$P < 0.01$.

**Table 3. Cardiac and vascular morphology**

| | Newborn | | | | 4-week-old | | | |
|---|---|---|---|---|---|---|---|---|
| | Control | FGR | FGR+MLT | P value | Control | FGR | FGR+MLT | P value |
| **Heart** | | | | | | | | |
| Globularity (AU g$^{-1}$) | 0.17 ± 0.04 | 0.24 ± 0.11 | 0.21 ± 0.05 | P = 0.0547 | 0.07 ± 0.01 | 0.1 ± 0.02 | 0.08 ± 0.02 | P = 0.1165 |
| Width (mm g$^{-1}$) | 1.17 ± 0.23 | 1.59 ± 0.46 | 1.42 ± 0.32 | P = 0.0863 | 0.72 ± 0.05 | 0.89 ± 0.19 | 0.78 ± 0.15 | P = 0.2283 |
| Length (mm g$^{-1}$) | 1.61 ± 0.36 | 2.17 ± 0.66 | 1.96 ± 0.39 | P = 0.0968 | 0.98 ± 0.07 | 1.18 ± 0.27 | 1.05 ± 0.24 | P = 0.3832 |
| Right ventricle wall width (mm g$^{-1}$) | 0.05 ± 0.02 | 0.06 ± 0.03 | 0.05 ± 0.02 | P = 0.4392 | 0.01 ± 0.00 | 0.01 ± 0.00 | 0.01 ± 0.00 | P = 0.4266 |
| Left ventricle wall width (mm g$^{-1}$) | 0.05 ± 0.03 | 0.06 ± 0.03 | 0.06 ± 0.02 | P = 0.3340 | 0.02 ± 0.00 | 0.02 ± 0.00 | 0.02 ± 0.00 | P = 0.2151 |
| Interventricular septal wall width (mm g$^{-1}$) | 0.06 ± 0.03 | 0.07 ± 0.03 | 0.07 ± 0.02 | P = 0.3828 | 0.02 ± 0.00 | 0.02 ± 0.00 | 0.02 ± 0.00 | P = 0.1514 |
| **Carotid** | | | | | | | | |
| % Area, intima-media | 60.8 ± 4.8 | 54.3 ± 7.6 | 57.1 ± 5.2 | P = 0.1930 | 64.1 ± 6.8 | 65.8 ± 6.0 | 64.9 ± 2.3 | P = 0.8341 |
| % Area, adventitia | 39.1 ± 4.8 | 45.7 ± 7.6 | 42.9 ± 5.2 | P = 0.1930 | 35.9 ± 6.8 | 34.2 ± 6.0 | 35.1 ± 2.3 | P = 0.8341 |
| Intima-media thickness (µm/mm$^2$) | 100.3 ± 13.2 | 95.9 ± 6.7 | 101.3 ± 17.5 | P = 0.7516 | 84.5 ± 13.5 | 94.6 ± 9.3 | 87.3 ± 8.9 | P = 0.2131 |
| % αSMA | 46.2 ± 3.1 | 37.9 ± 6.0* | 36.8 ± 7.4* | **P = 0.0240** | 46.5 ± 11.1 | 47.0 ± 8.4 | 50.8 ± 6.1 | P = 0.6049 |
| Elastin:Collagen (arbitrary units) | 0.8 ± 0.3 | 0.9 ± 0.6 | 0.9 ± 0.3 | P = 0.8767 | 0.8 ± 0.6 | 0.8 ± 0.4 | 0.9 ± 0.8 | P = 0.8974 |
| **Femoral** | | | | | | | | |
| % Area, intima-media | 48.7 ± 3.7 | 46.2 ± 5.2 | 46.6 ± 1.9 | P = 0.4679 | 48.0 ± 3.4 | 46.4 ± 1.7 | 48.7 ± 2.3 | P = 0.1792 |
| % Area, adventitia | 51.3 ± 3.7 | 53.8 ± 5.2 | 53.4 ± 1.9 | P = 0.4679 | 52.0 ± 3.4 | 53.6 ± 1.7 | 51.3 ± 2.3 | P = 0.1792 |
| Intima-media thickness (µm/mm$^2$) | 93.2 ± 3.9 | 81.6 ± 30.9 | 83.0 ± 13.7 | P = 0.7868 | 64.9 ± 16.4 | 79.3 ± 11.8 | 81.1 ± 12.8 | P = 0.0613 |
| % αSMA | 30.4 ± 1.7 | 28.3 ± 5.6 | 28.4 ± 3.7 | P = 0.5642 | 27.0 ± 3.1 | 23.9 ± 2.3 | 30.3 ± 3.4† | **P = 0.0027** |
| Elastin:Collagen (arbitrary units) | 1.0 ± 0.2 | 0.7 ± 0.2 | 1.2 ± 0.3† | **P = 0.0372** | 0.6 ± 0.1 | 0.9 ± 0.4 | 0.8 ± 0. | P = 0.1719 |

Mean ± SD. *P < 0.05, vs. control, †P < 0.05, vs. FGR. Abbreviation: αSMA, α-smooth muscle actin. Newborn: control (n = 6–9), FGR (n = 6–9), FGR+MLT (n = 4–7); 4-week-old: control (n = 5–9), FGR (n = 6–8), FGR+MLT (n = 7–8). Bold was used to indicate a p-value < 0.05.

increasing age ($P = 0.858$). This altered development was not ameliorated with maternal melatonin treatment. Analysis between groups showed that newborn FGR and FGR+MLT lambs had a reduced abundance of smooth muscle in the carotid artery ($P = 0.024$) (Table 3) compared to control lambs; however, this normalised by 4 weeks of age. By contrast, the amount of smooth muscle in femoral arteries of newborn lambs was not different between groups ($P = 0.564$) (Table 3), although FGR+MLT lambs showed a progressive increase in

smooth muscle compared to FGR lambs by 4 weeks of age ($P = 0.002$).

Vascular elastin-to-collagen ratio was assessed as a measure of stiffness, and neither the carotid nor femoral artery, the day after birth or at 4 weeks of age, was different in FGR lambs compared to controls (Table 3). Femoral artery elastin-to-collagen ratio was significantly increased in newborn FGR+MLT lambs compared to FGR lambs ($P = 0.033$), but not at 4 weeks of age.

### Biochemical markers in the tunica media

Markers of inflammation (CRP), oxidative stress (8-OHdG), angiogenesis (VEGF) and adrenergic receptors ($\alpha_{1A}$-AR) were assessed using immuno-histochemistry in the tunica media of carotid and femoral arteries (Fig. 4). Developmentally, the trajectory of carotid artery VEGF expression decreased 4.7-fold in FGR lambs from the day after birth to 4 weeks ($P = 0.041$) (Fig. 4*E* and *F*). By contrast, carotid VEGF expression in the FGR+MLT lambs significantly increased 3.7-fold between birth and 4 weeks ($P = 0.023$). As a result, VEGF abundance in the carotid arteries of 4-week-old FGR+MLT lambs was increased in comparison to FGR lambs ($P = 0.033$). Similarly, there was a treatment effect in the femoral artery, with *post hoc* analysis showing a trend towards increased VEGF in FGR+MLT 4-week-old lambs compared to controls ($P = 0.056$).

In carotid and femoral arteries, inflammatory CRP expression decreased with age in control ($P_{carotid} = 0.030$ and $P_{femoral} < 0.0001$) and FGR+MLT animals ($P_{carotid} = 0.005$ and $P_{femoral} = 0.007$, respectively) (Fig. 4*A* and *B*). FGR lambs also showed declining CRP in the femoral artery ($P = 0.0008$), but this was not observed in the carotid artery ($P = 0.061$). There was no difference in CRP expression across groups in either artery the day after birth ($P_{carotid} = 0.059$ and $P_{femoral} = 0.120$ but, by 4 weeks of age, CRP expression was elevated in the femoral arteries of FGR+MLT lambs compared to control ($P < 0.0001$) and FGR lambs ($P = 0.0009$) (Fig. 4*A* and *B*).

Vascular oxidative stress (8-OHdG) (Fig. 4*C* and *D*) became reduced over time in FGR and FGR+MLT lambs within the carotid artery ($P = 0.041$ and $P = 0.0008$), but the same effect was not observed in control lambs. Conversely, oxidative stress was reduced with age in the femoral artery of control lambs ($P = 0.004$). In the newborn age group, there were no differences in 8-OHdG expression in carotid or femoral arteries across groups. At 4 weeks, FGR+MLT femoral arteries had an increased abundance of 8-OHdG-positive nuclei in the tunica media relative to controls ($P = 0.025$).

The postsynaptic receptors $\alpha_{1A}$-AR are found within the smooth muscle of blood vessels and are key to

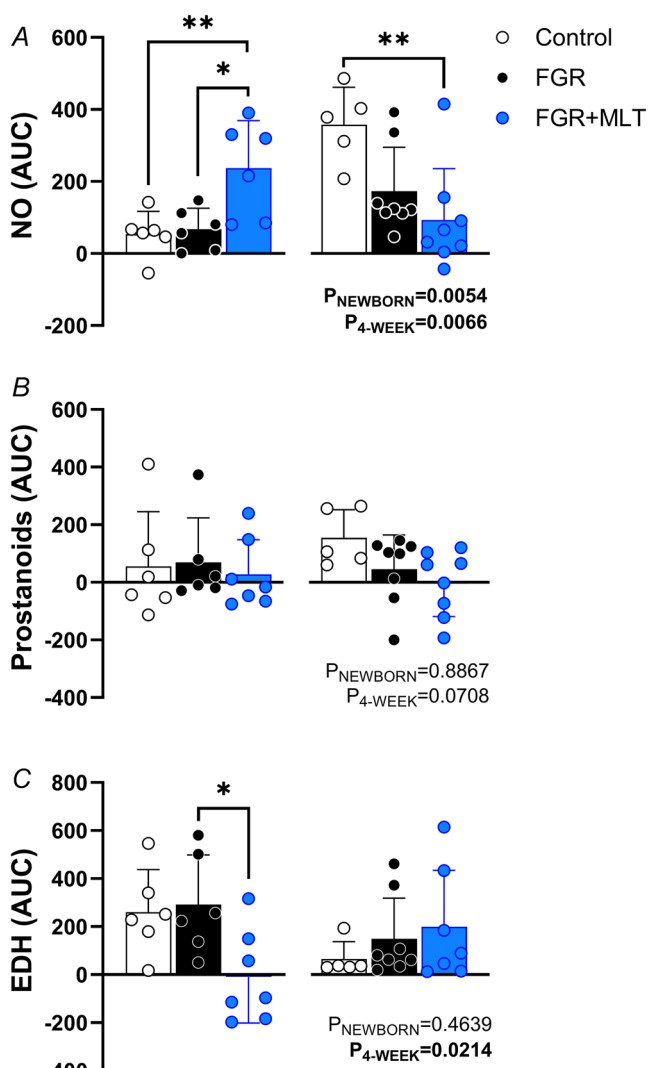

**Figure 3. Vasodilatation attributable to nitric oxide (NO), prostanoids and endothelium-derived hyperpolarisation (EDH)**
The NO- (*A*), prostanoid- (*B*) and EDH-dependent (*C*) components of vasodilatation in isolated third-order femoral resistance arteries, analysed using one-way ANOVA within age groups, with a *post hoc* Tukey test when applicable. Newborn: control ($n = 6$), FGR ($n = 5$), FGR+MLT ($n = 7$); 4-week-old: control ($n = 5$), FGR ($n = 8$) and FGR+MLT ($n = 8$). AUC, area under the curve. *$P < 0.05$, **$P < 0.01$.

the control of local tone through vasoconstriction. The population of $\alpha_{1A}$-AR in both the carotid artery and femoral artery significantly decreased with age in all treatment groups. However, it was observed that there was a greater difference in $\alpha_{1A}$-AR density in the control lambs over time (10- and 30-fold reduction in the carotid and femoral, respectively, $P < 0.0001$ and $P < 0.0001$) (Fig. 4*G*

and *H*). Melatonin-treated lambs also demonstrated an $\sim$11-fold reduction in $\alpha_{1A}$-AR in the carotid artery ($P < 0.0001$) but only a 3-fold decrease in the femoral artery ($P = 0.003$). $\alpha_{1A}$-AR abundance was greater in the femoral artery of 4-week-old FGR+MLT lambs compared to control lambs ($P = 0.006$). An examination of tyrosine hydroxylase staining was undertaken to evaluate vascular

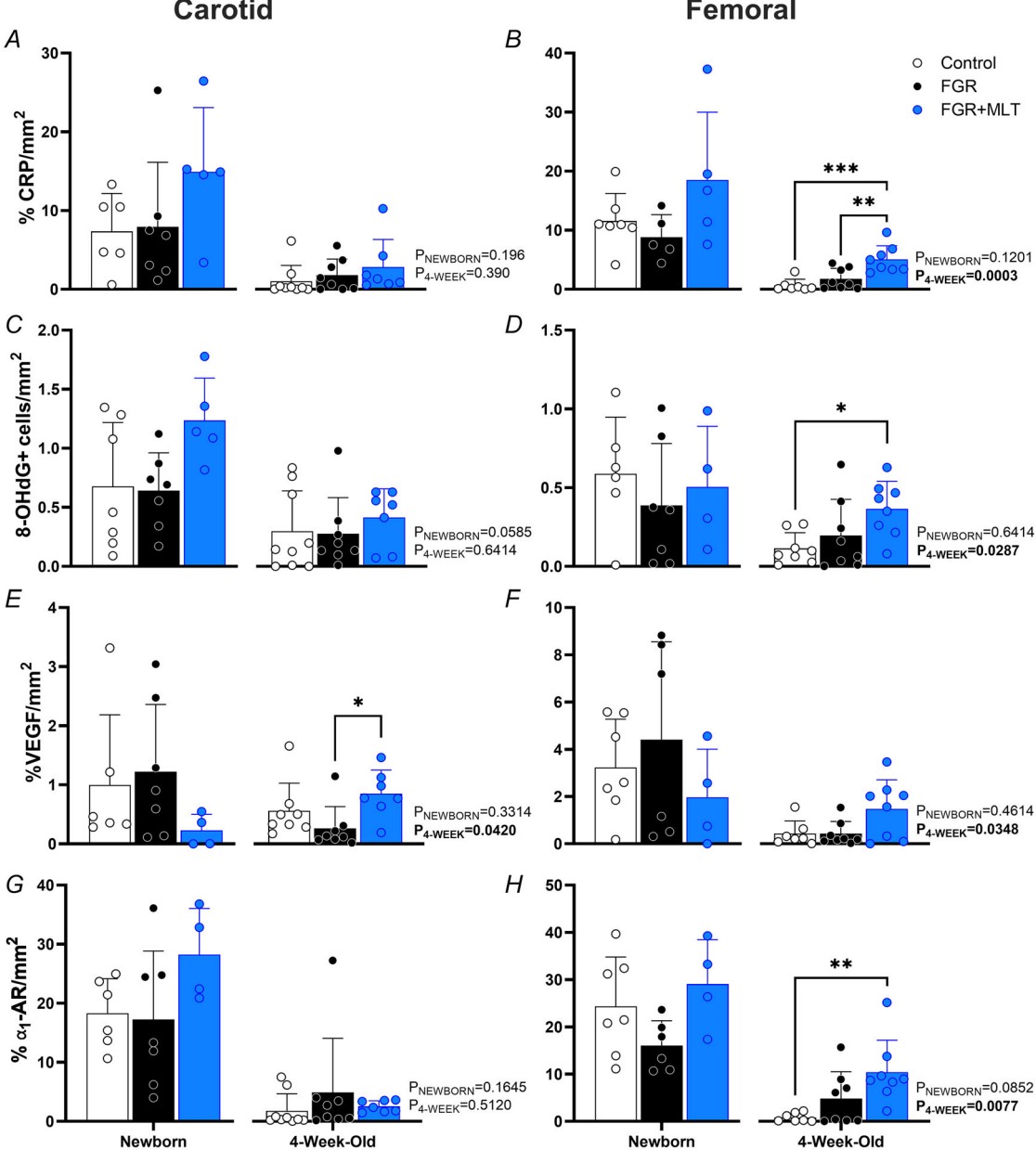

**Figure 4. Quantification of biochemical markers in the carotid and femoral arteries**
Percentage area positive staining of C-reactive protein (CRP) in the carotid artery (%) (*A*) and the femoral artery (%) (*B*); number of 8-hydroxy-2′-deoxyguanosine (8-OHdG) positive cells in the carotid artery (*C*) and the femoral artery (*D*); percentage area positive staining of vascular endothelial growth factor (VEGF) in the carotid artery (%) (*E*) and femoral artery (%) (*F*); and percentage area positive staining $\alpha_1$-adrengeric receptors ($\alpha_1$-AR) in the carotid artery (%) (*G*) and femoral artery (%) (*H*). Data were analysed using a one-way ANOVA within age groups, with a *post hoc* Tukey test when applicable. Newborn: control ($n = 6$–7), FGR ($n = 5$–7), FGR+MLT ($n = 4$–5); 4-week-old: control ($n = 7$–9), FGR ($n = 8$) and FGR+MLT ($n = 8$). *$P < 0.05$, **$P < 0.01$, ***$P < 0.001$.

sympathetic innervation. There was no difference in tyrosine hydroxylase expression at either age or across groups (carotid: newborn control $0.7 \pm 1.4\%$ *vs.* newborn FGR $3.2 \pm 6.5\%$ *vs.* newborn FGR+MLT $7.3 \pm 12.4\%$, $P = 0.3519$; 4-week-old control $0.1 \pm 0.1\%$ *vs.* 4-week-old FGR $0.1 \pm 0.2\%$ *vs.* 4-week-old FGR+MLT $1.8 \pm 3.7\%$, $P = 0.2095$. Femoral: newborn control $5.2 \pm 7.8\%$ *vs.* newborn FGR $3.7 \pm 4.3\%$ vs. newborn FGR+MLT $1.4 \pm 1.4\%$, $P = 0.5816$; 4-week-old control $0.2 \pm 0.2\%$ *vs.* 4-week-old FGR $1.4 \pm 2.7\%$ *vs.* 4-week-old FGR+MLT $4.1 \pm 8.0\%$, $P = 0.345$).

## Discussion

Chronic hypoxia *in utero* and FGR alters cardiovascular development and, in turn, increases lifelong risk for cardiovascular disease. Here, the hypothesis was tested that melatonin would prevent the developmental programming of structural and functional deficits in the vasculature of postnatal growth-restricted lambs. This study demonstrates that antenatal maternal melatonin attenuates the development of peripheral vascular endothelial dysfunction in 4-week-old FGR lambs, as vasodilatory capacity improved between the day after birth and 4 weeks of age in FGR+MLT lambs but declined in untreated FGR lambs. Antenatal melatonin exposure did not improve fetal growth; however, at 4 weeks of age, the FGR+MLT lambs no longer demonstrated a deficit in weight relative to control lambs at the same age. Interestingly, antenatal melatonin treatment reduced the brain-sparing response, with newborn FGR lambs treated with melatonin demonstrating significant, but symmetric, growth restriction on the day after birth. It was also noted that, by 4 weeks of age, lambs exposed to antenatal melatonin more probably had elevated levels of vascular inflammation and oxidative stress in the periphery, which requires consideration.

Previous clinical and preclinical research comprehensively shows that FGR alters the extracellular matrix composition in major conduit arteries, leading to stiffer, less compliant vessels, which increases resistance and reduces overall system efficiency (Rock et al., 2023; Sehgal et al., 2018, 2019, 2023). This developmental profile of cardiovascular change programs for long-term poor cardiovascular health, resulting in elevated blood pressure from early childhood, which subsequently increases the risk of developing hypertension later in life. In the present study, we did not observe differences in basal blood pressure, consistent with previous ovine models of FGR (Danielson et al., 2005; Edwards et al., 1999) but in contrast to clinical studies in neonates (Sehgal et al., 2018), children and adults (Huxley et al., 2000). However, our findings provide novel insight that FGR causes subtle changes in cardiovascular structure and

function that are modified over the first postnatal weeks, highlighting the importance of multiple assessment timepoints. We detected a reduction in the abundance of smooth muscle in the carotid artery of newborn FGR lambs and endothelial dysfunction in the femoral artery at 4 weeks of age. These findings highlight the progressive nature of vascular changes associated with FGR. These deficits probably contribute to altered vascular function in the immediate postnatal period and elevate the risk of developing cardiovascular disease in adulthood. Melatonin treatment did not ameliorate smooth muscle deficits in carotid vessels of FGR lambs, but, at 4 weeks of age, femoral arteries from melatonin-treated lambs had a significant increase in smooth muscle abundance, indicating differential effects on carotid and femoral vasculature development.

The brain-sparing phenotype that is typically seen in our model of SUAL-induced FGR and, in clinical FGR, was blunted in the FGR lambs exposed to melatonin. Previous studies have shown that melatonin does not improve uterine artery blood flow in models of FGR; however, it increases umbilical artery blood flow (Lemley et al., 2011; Thakor et al., 2010) probably via increasing cardiac output through the descending aorta. The mechanisms regulating the brain-sparing response to *in utero* hypoxia have been explored, combining robust peripheral vasoconstriction with central vasodilatation responses (Allison et al., 2016). Interestingly, although our FGR+MLT lambs were growth-restricted, the growth restriction was symmetrical, indicating that cardiac output was not preferentially distributed to the brain with melatonin treatment. Thakor et al. (2015) previously showed a lack of brain-sparing response to acute hypoxia when undertaken during melatonin exposure *in vivo*. In their study, it was demonstrated that peripheral vasoconstriction was abolished and this was, at least in part, mediated by increased NO bioavailability in the periphery (Thakor et al., 2015). It is likely, therefore, that chronic exposure to antenatal melatonin dampened the hypoxia-induced brain sparing in FGR via melatonin-induced peripheral vasodilatation in a similar manner. Chronic peripheral vasoconstriction is a key contributor to the development of endothelial dysfunction, and our assessment of femoral endothelium-dependent NO vasodilatation, along with other researchers in our field, has shown that dampening this vasoconstriction by improving NO bioavailability production and/or consumption prevents this cardiovascular dysfunction.

Melatonin can also alter vascular tone via melatonin receptors (MT) and the specific receptor actions vary between vascular beds. Melatonin's vaso-modulatory effects are mediated through three receptor subtypes: $MT_1$, which promotes vasodilatation; $MT_2$, which promotes vasoconstriction; and $MT_3$, which contributes to tissue detoxification via antioxidant mechanisms. These

receptor subtypes are expressed throughout the body, including in the vasculature and the brain (Malhotra et al., 2024). The relative abundance of receptor populations within organs probably dictates the differential impact of melatonin on organ-specific blood flow. However, the distribution of MT receptors in development and beyond, as well as their physiological roles in the vasculature, have not been thoroughly explored. Cook et al. (2011) and van der Helm-van Mil et al. (2003) demonstrated that oral melatonin (3 mg) or an i.v. melatonin bolus (10 mg), respectively, did not alter heart rate, mean arterial pressure or cerebral blood flow. In the cerebral vascular bed, the dampening of the brain-sparing response could be mediated by receptor-mediated changes in blood flow (Cook et al., 2011; van der Helm-van Mil et al., 2003). MT receptor populations are present within FGR fetal lamb brains, and elevated by FGR in a region-specific manner (Malhotra et al., 2024), but we have not yet characterised MT receptor distribution within the vasculature. The implications of modifying or abolishing the brain-sparing response through peripheral vasodilatation are not known, but are certainly of interest (Piscopo et al., 2025). Melatonin is shown to have protective benefits for the FGR brain in fetal and newborn sheep, by increasing myelination and preventing axonal damage (Malhotra et al., 2024; Miller et al., 2014), and whether this improves the trajectory of brain development in FGR (Dudink et al., 2025) needs further consideration.

The sympathetic arm of the autonomic nervous system is critical for the maintenance of blood pressure and, importantly, in the context of FGR, mediating the brain-sparing response via peripheral vasoconstriction.(Giussani, 2016) Previous preclinical studies have shown that FGR alters components of the sympathetic nervous system, including hyperinnervation of peripheral vascular beds of hypoxic chicken embryos (Ruijtenbeek et al., 2000) and increased plasma catecholamine concentration in growth-restricted fetal sheep (Simonetta et al., 1997). Therefore, components of the sympathetic nervous system that mediate blood pressure regulation were interrogated to determine potential implications of melatonin administration. In the present study, all lambs were normotensive. This is in keeping with studies using the carunclectomy model of ovine FGR that have demonstrated that growth-restricted fetuses are normotensive at rest, but then respond with a greater decrease in blood pressure when $\alpha_{1A}$-AR activity is blocked with phentolamine compared to control fetuses (Danielson et al., 2005; Darby et al., 2021). In the present study, the abundance of $\alpha_{1A}$-AR was not different between control and FGR lambs at either timepoint and both groups showed a pruning of $\alpha_{1A}$-AR with age. By contrast, $\alpha_{1A}$-AR abundance in the femoral artery was elevated in normotensive 4-week-old FGR+MLT lambs. Combined, these findings suggest that in FGR

fetuses exposed to melatonin treatment, there is a greater reliance in $\alpha_{1A}$-adrenergic activation for blood pressure regulation at 4 weeks of age. Activation of $\alpha_{1A}$-AR occurs primarily with the binding of catecholamines, which are released systemically by the adrenal medulla or locally by postganglionic neurons in the target artery to elicit vasoconstriction (Perez, 2021). Circulating catecholamine concentrations were not assessed in the present study; however, a previous study showed that circulating adrenaline and noradrenaline levels are not different between FGR and control lambs in our model (Rock et al., 2023). Furthermore, local differences in vascular tyrosine hydroxylase expression, the rate-limiting enzyme for catecholamine synthesis, were not observed in FGR+MLT lambs. It would be useful to determine whether this increase in $\alpha_{1A}$-AR density persists into adulthood, or alternatively, whether melatonin delays the maturational pruning of the receptors, with an eventual decrease to control levels but at a slower rate.

The immediate newborn period presents a transition from a low-oxygen *in utero* environment to an oxygen-rich external environment, with an upregulation of oxygen availability probably generating an excess of ROS, causing oxidative stress. In the present study, the oxidative stress marker, MDA, was elevated in FGR lambs the day after birth. This finding is not surprising given the ample research showing the role of oxidative stress in the pathology of FGR. Circulating melatonin levels in FGR+MLT lambs remained elevated the day after birth compared to control and FGR lambs, aligning with baseline values reported in other studies (Tare et al., 2014). By contrast to expectations and previous findings, melatonin exposure did not normalise the elevated circulating oxidative stress observed in FGR lambs. Although melatonin is frequently characterised as an antioxidant, direct evidence for its ROS-scavenging activity *in vivo* remains limited, and its vascular effects may be predominantly receptor-mediated (Liu et al., 2016; Monteiro et al., 2024; Paulis et al., 2012). The chemical reactivity of superoxide with NO far outpaces its potential interaction with melatonin (Huie & Padmaja, 1993; Zang et al., 1998). Instead, melatonin appears to exert a stronger effect by enhancing endogenous antioxidant systems, particularly through the upregulation of superoxide dismutase, which neutralises superoxide more efficiently than melatonin itself (Morvaridzadeh et al., 2020). This distinction may help to explain our findings in FGR+MLT lambs, where melatonin treatment only partially reduced oxidative stress. The effect is consistent with reinforcement of antioxidant defences rather than direct actions of melatonin on the elimination of ROS. An alternative explanation for the observed results may be the phenomenon of 'rebound oxidative stress,' previously described in studies involving antioxidant withdrawal (Surikow et al., 2018). This transient response may stem

from the abrupt discontinuation of exogenous melatonin coupled with increased blood oxygenation after birth.

Newborn FGR+MLT lambs displayed evidence of endothelial dysfunction, with reduced endothelium-dependent vasodilatory capacity in the femoral artery. This was unexpected because a previous study using a similar study design demonstrated that antenatal melatonin improved endothelium-dependent vasodilatory capacity in coronary arteries from newborn lambs (Tare et al., 2014). However, there are also significant differences between these studies, including the dose of melatonin (6 mg/day in the previous study *vs.* 15 mg day$^{-1}$ in the present study), the vascular beds examined (which in the earlier study were likely not subjected to chronic *in utero* vasoconstriction), and the timing of treatment initiation (0.7 gestation previously *vs.* 0.6 gestation in our study) (Tare et al., 2014). To the best of our knowledge, our study is the first to assess peripheral vascular function in newborns exposed to antenatal melatonin (Rock et al., 2024) and the observation of perinatal endothelial dysfunction represents a novel finding. In the present study, the dose of melatonin was selected to be pharmacologically comparable to the PROTECTMe clinical trial (Palmer et al., 2019) (30 mg orally *vs.* 15 mg i.v.), supporting the clinical relevance of our exposure paradigm. The discrepancies in vascular outcomes between the present study and previous studies underscore the complexity of melatonin's effects on the cardiovascular system. Endothelial dysfunction during the perinatal period may be particularly detrimental, as this is a time of cardiovascular adaptation, a risk heightened in growth-restricted neonates (Oyang et al., 2023). The observed endothelial dysfunction in newborn FGR+MLT lambs was transient, with endothelial function improving by 4 weeks of age. Interestingly, when statins, which also have antioxidant properties, are used to alleviate endothelial dysfunction and treatment is withdrawn, mice display transient endothelial dysfunction lasting up to 1 week as a result of elevated superoxide formation (Vecchione & Brandes, 2002). The potential risks of rebound oxidative stress or transient endothelial changes highlight the need for close monitoring of oxidative and vascular markers during and after treatment.

Conversely, FGR lambs demonstrated a progressive decline in endothelial function that is probably the result of an increase in cyclooxygenase-2 (COX-2) in response to chronic fetal hypoxia, programming for a reduction in the contribution of the prostanoid pathway to vasodilatation in FGR lambs postnatally (Rock et al., 2023). When metabolised, melatonin forms $N^1$-acetyl-$N^2$-formyl-5-methoxykynuramine and $N^1$-acetyl-5-methoxykynuramine, which are COX-2 inhibitors and form part of melatonin's anti-inflammatory profile; thus, melatonin may also modulate COX-2 during gestation and prevent the developmental programming of

endothelial dysfunction. It has also been highlighted that antioxidant treatments such as melatonin can improve NO bioavailability in the fetal vasculature in pregnancies complicated with FGR and prevent endothelial dysfunction. The present study demonstrates that newborn FGR+MLT lambs had improved NO handling in the femoral artery compared to both control and FGR lambs. However, this occurred concomitantly with a reduction in EDH contribution to vasodilatation and an overall reduced ability to vasodilate in the newborn period. Despite the fact that the contribution of NO to endothelium-dependent vasodilatation was improved in newborn FGR+MLT lambs, it was insufficient to improve overall vasodilatory capacity in lambs examined the day after birth. This reflects a shift in endothelial control of vascular tone, with greater reliance on NO-dependent mechanisms. Additionally, melatonin has been shown to inhibit voltage-gated potassium channels in peripheral arteries of rodents (Lew & Flanders, 1999), crucial for EDH-mediated responses, contributing to the reduction in EDH contribution following melatonin treatment. Consequently, overall endothelial vasodilatation remains impaired because the enhancement of NO activity was insufficient to offset the loss of EDH-mediated vasodilatation. Furthermore, the increase in NO endothelium-dependent vasodilatation in newborn FGR+MLT lambs did not persist, with NO contribution to vasodilatation being significantly impaired in 4-week-old FGR+MLT lambs. Collectively, the data show that, although intervention with antenatal maternal melatonin improves endothelial function in the longer term, this is not mediated by a sustained improvement in the contribution of NO to endothelium-dependent vasodilatation.

Oxidative stress and inflammatory markers (8-OHdG and CRP, respectively) were assessed in both carotid and femoral arteries of newborn and 4-week-old lambs. The most notable result from this analysis is that 4-week-old FGR+MLT lambs had increased indices of oxidative stress and inflammation in the femoral artery. Oxidative stress and inflammation can upregulate the transcription factor, hypoxia-inducible factor-1$\alpha$ (HIF-1$\alpha$). Although HIF-1$\alpha$ was not directly investigated, immunohistochemical analysis demonstrated an increase in VEGF, one of the downstream signalling pathways of HIF-1$\alpha$, in both the femoral and carotid arteries. Melatonin has been shown to suppress HIF-1$\alpha$ and, subsequently, VEGF, in endothelial cells. Therefore, the changes seen in the present study are consistent with a rebound effect following melatonin withdrawal. Further research into whether this rebound effect persists past 4 weeks of age would be informative to comprehensively determine the clinical utility of antenatal maternal melatonin administration on the cardiovascular system long-term. Inflammation is widely recognised as a pre-

cursor to cardiovascular disease, with elevated levels of CRP associated with an increased risk of cardiovascular events (Sakkinen et al., 2002). In our study, antenatal melatonin reduced indices of oxidative stress, yet was associated with persistently high vascular CRP at 4 weeks after birth. This paradoxical finding is novel and warrants further longer-term follow-up, given that elevated CRP is a biomarker of vascular inflammation and a predictor of adult-onset atherosclerosis and cardiovascular disease (Frary et al., 2020). In rodent models, chronic melatonin supplementation has been reported to improve long-term vascular reactivity (Hansell et al., 2022), but whether these benefits are also seen in larger-animal models and humans or persist when treatment ceases at birth remains unknown. Accordingly, the implications of sustained vascular inflammation in FGR+MLT lambs remain unresolved and requires further study into adolescence and adulthood. It would also be valuable to assess whether a tapered ante- or postnatal withdrawal of melatonin would be beneficial to prevent the negative vascular outcomes observed in the present study. Melatonin is a well-established neuroprotective agent in perinatal compromise, including for FGR, with melatonin currently under investigation in human pregnancies complicated by FGR in the PROTECTMe trial (Palmer et al., 2019). Our findings highlight the importance of extending such assessments beyond the brain to include postnatal cardio-vascular end-points. FGR affects multiple organ systems, with pathology evident in the cardiovascular system. Although our findings support potential cardiovascular benefits, they also highlight the need for careful evaluation before clinical translation can be recommended.

## Limitations

We acknowledge the limitations of the present study. Our study did not include a control group of lambs exposed to melatonin; this choice was based on a clinical intervention that would only be applied to FGR pregnancies. Melatonin was diluted in ethanol (1% in saline) and, although this concentration is low, acute infusion of ethanol has been shown to cause vasoconstriction at rest and augment the effect of vasodilators. Given the complexity and long-term nature of this study, control and FGR lambs did not receive an ethanol control vehicle, which should be taken into consideration when interpreting these results. Clinically, melatonin is administered as an oral slow-release tablet (Palmer et al., 2019), thus removing the need for ethanol. Furthermore, melatonin regulates circadian rhythm and is used as a sleep aid (Yiallourou et al., 2016) and, although anecdotally, we did not observe changes in sleep patterns in the melatonin-treated animals, we did not formally assess sleep patterns in the mum or lambs, and high-dose

melatonin to newborn lambs does increase their time spent sleeping (Aridas et al., 2018, 2021).

An important limitation to acknowledge is that only blood vessels excised from one vascular bed could be studied because of practical constraints. After careful consideration, it was decided that functional vascular data from the femoral artery would be valuable, especially because peripheral vascular beds significantly contribute to hypertension, a major health issue for adults born growth-restricted. Inclusion of additional vascular beds, such as the carotid or middle cerebral artery, would have provided a more comprehensive assessment of the impact of FGR and melatonin on the vasculature. Although every possible effort was made to evenly distribute sexes between the groups, it was not wholly possible to control for sex differences, as every viable pregnancy allocated to this project was used.

The present study is among the first to demonstrate the postnatal cardiovascular implications of antenatal melatonin administration, with assessments conducted at multiple postnatal timepoints. Our findings in the femoral artery raise important questions around the potential for rebound inflammatory or oxidative stress following the withdrawal of exogenous melatonin at birth. Further work is needed to define the pathways that drive these changes in inflammation and oxidative stress, and whether vascular inflammation at 4 weeks persists to adulthood, together with the cardiovascular implications.

## Future directions

The results of the present study indicate that melatonin has the potential to mitigate the progressive development of impaired endothelium-dependent vasodilatation in growth-restricted lambs. However, this benefit is associated with transient impairment of endothelial function on the first day of life. It would be prudent to investigate the physiological implications of this early vascular dysfunction during the critical period of cardio-vascular adaptation to postnatal life. Adjustments to the antenatal melatonin treatment regime, such as tapering the dose before and after birth, could be explored to support a smoother cardiovascular transition on the first day of life.

Additionally, because the present study was conducted up to 4 weeks of age, equivalent to an ∼1-year-old human infant, extending the study outcomes into adulthood is important to determine whether endothelial function is sustained or continues to improve with age, potentially achieving full restoration to the relaxation abilities observed in control lambs. Furthermore, given that functional assessment of the carotid artery was not possible in our study, future research should include such

testing to provide a more comprehensive understanding of how FGR impacts vascular function.

The present study provides novel insight into the short- and longer-term, and region-specific impact of melatonin on the cardiovascular system. Our findings demonstrate that, although antenatal melatonin improves the contribution of NO to endothelium-dependent vaso-dilatation in the femoral artery of newborn FGR+MLT lambs, it is accompanied by an overall reduction in femoral endothelium-dependent vasodilatory capacity. Notably, this reduction in endothelial function is trans-ient and improves by 4 weeks of age, which contra-sts with the progressive impairment of endothelial function seen in untreated FGR lambs. Immuno-histochemical analysis revealed elevated oxidative stress and inflammatory markers in the femoral artery of 4-week-old FGR+MLT lambs. This differential profile of cardiovascular development after antenatal melatonin exposure is probably a result of a modification of the brain-sparing response secondary to peripheral vaso-dilatation and concomitant removal of melatonin at birth with an increase in circulating oxygen concentration. Thus, our findings demonstrate that melatonin exerts both beneficial and adverse effects on the developing cardiovascular system in growth-restricted lambs. This highlights the need for further research to evaluate whether melatonin can confer further long-term cardio-vascular benefits in growth-restricted neonates and to ascertain whether the observed adverse effects outweigh the positive outcomes.

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

## Additional information

### Data availability statement

All data supporting the findings of this study are available within the published paper and are available from the corresponding author upon reasonable request.

### Competing interests

The authors declare that they have no competing interests.

### Author contributions

C.R.R., S.L.M. and B.J.A. conceived and designed research. C.R.R., T.A.W., B.R.P., A.E.S., Y.P., C.K., F.L.C., S.L.M. and B.J.A. performed experiments. C.R.R. analysed data. C.R.R., S.L.M. and B.J.A. interpreted the results of experiments. C.R.R. prepared figures. C.R.R., S.L.M. and B.J.A. drafted the manuscript. C.R.R., T.A.W., B.R.P., A.E.S., Y.P., C.K., E.J.C., F.L.C., S.L.M. and B.J.A. edited and revised the manuscript. C.R.R., T.A.W., B.R.P., A.E.S., Y.P., C.K., E.J.C., F.L.C., S.L.M. and B.J.A. approved the final version of the manuscript submitted for publication.

### Funding

This project was supported by funding from the National Health and Medical Research Council Australia (Investigator Grants BJA #1175843; SLM #2016688 and Project Grant #1160393). The Monash University Faculty of Medicine, Nursing and Health Sciences Graduate Research Scholarship also supported this study. The funding bodies had no input in the design of the study and collection, analysis and interpretation of data.

### Acknowledgements

We thank the animal house staff and staff at the Monash Animal Research Platform for their help with the experiments and care of the animals. We acknowledge the use of equipment and

technical assistance made available by the Monash Histology Platform, Department of Anatomy and Developmental Biology, Monash University.

## Keywords

brain sparing, cardiovascular, fetal growth restriction, melatonin

## Supporting information

Additional supporting information can be found online in the Supporting Information section at the end of the HTML view of the article. Supporting information files available:

**Peer Review History**

