## [Peer Review History · The Journal of Physiology]

Antenatal melatonin for cardiovascular deficits in fetal growth restriction

Charmaine R Rock, Tegan A White, Beth R Piscopo, Amy E Sutherland, Yen Pham, Emily J Camm, Connor Karozis, Fiona L Cousins, Suzanne L Miller, and Beth J Allison

DOI: 10.1113/JP288750

Corresponding author(s): Charmaine Rock (charmaine.rock@monash.edu)

The following individual(s) involved in review of this submission have agreed to reveal their identity: Bernardo J Krause (Referee #2)

Review Timeline:	Submission Date:	16-Feb-2025
	Editorial Decision:	09-Apr-2025
	Revision Received:	06-Oct-2025
	Editorial Decision:	20-Oct-2025
	Revision Received:	21-Oct-2025
	Accepted:	27-Oct-2025

Senior Editor: Laura Bennet

Reviewing Editor: Janna Morrison

Transaction Report:

Dear Dr Rock,

Re: JP-RP-2025-288750 "Antenatal melatonin for cardiovascular deficits in fetal growth restriction" by Charmaine R Rock, Tegan A White, Beth R Piscopo, Amy E Sutherland, Yen Pham, Emily J Camm, Connor Karozis, Fiona L Cousins, Suzanne L Miller, and Beth J Allison

Thank you for submitting your manuscript to The Journal of Physiology. It has been assessed by a Reviewing Editor and by 2 expert referees and we are pleased to tell you that it is potentially acceptable for publication following satisfactory major revision.

LANGUAGE EDITING AND SUPPORT FOR PUBLICATION: If you would like help with English language editing, or other article preparation support, Wiley Editing Services offers expert help, including English Language Editing, as well as translation, manuscript formatting, and figure formatting at www.wileyauthors.com/eoo/preparation. You can also find resources for Preparing Your Article for general guidance about writing and preparing your manuscript at www.wileyauthors.com/eoo/prepresources.

REVISION CHECKLIST:

We look forward to receiving your revised submission.

Yours sincerely,

Laura Bennet
Senior Editor
The Journal of Physiology

REQUIRED ITEMS

- Author photo and profile. First or joint first authors are asked to provide a short biography (no more than 100 words for one author or 150 words in total for joint first authors) and a portrait photograph. These should be uploaded and clearly labelled together in a Word document with the revised version of the manuscript. See Information for Authors for further details.

- The contact information for the person responsible for 'Research Governance' at your institution needs to be provided. This includes their name and an institutional email address. Please ensure the contact is not an author on this paper and provide an alternate contact if necessary, or confirm in the submission form that the author whose email was provided has sole responsibility for research governance. This is the person who is responsible for regulations, principles and standards of good practice in research carried out at the institution, for instance the ethical treatment of animals, the keeping of proper experimental records or the reporting of results.

- You must start the Methods section with a paragraph headed Ethical approval (https://jp.msubmit.net/cgi-bin/main.plex?form_type=display_requirements#methods).

Research must comply with The Journal's policies regarding animal experiments (<https://physoc.onlinelibrary.wiley.com/hub/animal-experiments>) and adherence to these policies must be stated in the manuscript.

Authors should confirm in their Methods section that their experiments were carried out according to the guidelines laid down by their institution's animal welfare committee, including an ethics approval reference number. The Methods section must contain a statement about access to food, water and housing, details of the anaesthetic regime: anaesthetic used, dose and route of administration, and method of killing the experimental animals.

- The reference list must be in alphabetical order, rather than numbered, to comply with our Journal format.

- Your paper contains Supporting Information of a type that we no longer publish, including supplementary tables and figures. Any information essential to an understanding of the paper must be included as part of the main manuscript and figures. The only Supporting Information that we publish are video and audio, 3D structures, program codes and large data files. Your revised paper will be returned to you if it does not adhere to our Supporting Information Guidelines.

- Papers must comply with the Statistics Policy: https://jp.msubmit.net/cgi-bin/main.plex?form_type=display_requirements#statistics.

In summary:

- If $n \leq 30$, all data points must be plotted in the figure in a way that reveals their range and distribution. A bar graph with data points overlaid, a box and whisker plot or a violin plot (preferably with data points included) are acceptable formats.
 - If $n > 30$, then the entire raw dataset must be made available either as supporting information, or hosted on a not-for-profit repository, e.g. FigShare, with access details provided in the manuscript.
 - 'n' clearly defined (e.g. x cells from y slices in z animals) in the Methods. Authors should be mindful of pseudoreplication.
 - All relevant 'n' values must be clearly stated in the main text, figures and tables.
 - The most appropriate summary statistic (e.g. mean or median and standard deviation) must be used. Standard Error of the Mean (SEM) alone is not permitted.
 - Exact p values must be stated. Authors must not use 'greater than' or 'less than'. Exact p values must be stated to three significant figures even when 'no statistical significance' is claimed.
- Please include an Abstract Figure file, as well as the Figure Legend text within the main article file. The Abstract Figure is a piece of artwork designed to give readers an immediate understanding of the research and should summarise the main conclusions. If possible, the image should be easily 'readable' from left to right or top to bottom. It should show the physiological relevance of the manuscript so readers can assess the importance and content of its findings. Abstract Figures should not merely recapitulate other figures in the manuscript. Please try to keep the diagram as simple as possible and without superfluous information that may distract from the main conclusion(s). Abstract Figures must be provided by authors no later than the revised manuscript stage and should be uploaded as a separate file during online submission labelled as File Type 'Abstract Figure'. Please also ensure that you include the figure legend in the main article file. All Abstract Figures should be created using BioRender. Authors should use The Journal's premium BioRender account to export high-resolution images. Details on how to use and access the premium account are included as part of this email.

EDITOR COMMENTS

Reviewing Editor:

Thank you for submitting this interesting paper.

The supplementary data suggests that t tests and on way ANOVAs were used to analyse the data. It is not clear why a 2 way ANOVA was not used.

All data points should be shown as well as the mean and SD.

The methods do not include information about access to food/water, housing or pain relief.

The postmortem describes collection of blood vessels but not the heart.

JP does not have supplementary data.

The Discussion would benefit from consideration of previous literature in the field of FGR, blood pressure regulation and SNS from authors such as Ruitjenbeek, Danielson, Zhang, Darby, Gilbert, Phillips (DIABETIC MEDICINE, 1997; 14: 673-677).

Other studies of FGR in sheep have also shown no change in blood pressure.

Line 555 refers to sleep patterns but these are not reported in the paper.

Senior Editor:

Thank you for your paper. Please address all reviewer comments fully for the paper to progress with The Journal and in particular those of reviewer two who raises many concerns. Please ensure you review our statistics and ethics policies to ensure the paper complies and provide all required methods.

REFEREE COMMENTS

Referee #1:

This timely manuscript by Rock and colleagues on the use of antenatal melatonin use during pregnancy has important implications on clinical use as melatonin continues to be a treatment of great interest. The article acknowledges both positive and negative impacts of treatment and adds to the growing evidence for specific targeted usage. The symmetrical growth restriction observed was of particular interest.

The manuscript is generally well written, this reviewer recommends some minor revisions that could strengthen the output.

1) I may have missed it, but I don't believe animal numbers were mentioned anywhere in the manuscript. In addition most of the graphs are simple bar graphs without individual data points making it difficult to gauge the power of the analyses.

2) Sex effects were mentioned in the discussion/limitation section, however, again this was not clarified in the methods the ratio of males to females within each group. This is particularly important as many studies have reported sex specific responses to melatonin treatment.

3) The Journal of Physiology policy is to use Standard deviation, not SEM unless properly justified.

4) I don't think the weight gain measure at day1 provides any meaningful purpose in the study as outlined in the discussion section. It would be more meaningful if all animals were measured 24hrs after birth.

Minor comments:

1) Line 132: Antibiotics were mentioned in the methods, but no detail was provided on what was given.

2) Line 241: I believe this should say 'non-parametric' here when describing how data that was not normally distributed was analyzed.

3) Line 244: "FGR lambs less than control lambs" missing word? Weighed?

4) Line 248: Missing P value "was significantly lower than that of controls in the same cohort."

5) Line 255: again, missing P-value at the end of the sentence.

6) Line 387: I think you mean symmetric not asymmetric here, as this is what happens in regular growth restriction.

7) The thickness of the error bars in the "Maximal response to SNP" graph is smaller than the other graphs. Would it be possible to include individual data points on all graphs like in Figure 4? It helps with transparency. Bar graphs can be used to 'hide' variability in data.

Referee #2:

The manuscript presents a comprehensive study of the effects of antenatal melatonin treatment on fetal growth restriction (FGR) and its implications for cardiovascular development in lambs. In this regard, it thoroughly examines the effects of melatonin on vascular function, oxidative stress, and inflammation in FGR lambs, providing data concerning the mechanisms underlying the observed changes, such as the role of melatonin receptors, NO bioavailability, and oxidative stress pathways. Furthermore, the study highlights the differential effects of melatonin on carotid and femoral vasculature, which adds depth to the understanding of region-specific vascular responses.

However, some major comments must be addressed:

Despite consistently being described as a scavenger with antioxidant effects, direct evidence concerning this trait for melatonin is elusive. Consequently, its vascular effects potentially involve mainly receptor-mediated actions rather than direct targeting of oxidant agents (PMID: 22916799; 26514204; 21752095). This issue must be acknowledged in the introduction and discussion.

Myography results must be informed as most studies do. For instance, the maximal relaxing response is the difference between the active tone after and before adding the relaxing agent, but not the residual tone after stimulation. Please refer to

more standardized protocols available if required (PMID: 16028678; 33989082).

Considering the low number of subjects studied, data in tables must be informed as the median and interquartile range, and statistics performed considering non-parametric test (PMID: 27556235). Similarly, individual values must be informed in figures 1 - 3, as shows figure 4.

The study provides detailed mechanistic insights, but some findings remain inconsistent or unexplained. For instance, melatonin improved NO response in newborn FGR lambs but did not improve overall vasodilatory capacity. The reasons for this are not fully addressed. Similarly, the transient endothelial dysfunction in newborn FGR+MLT lambs differs from previous studies showing improved vasodilation. Some potential reasons for these differences, such as dosage or timing of melatonin administration need to be explored.

While the study focuses on outcomes at 24 hours and 4 weeks of age, the discussion does not sufficiently address the long-term implications of the observed changes. Will the elevated oxidative stress and inflammation in 4-week-old FGR+MLT lambs persist into adulthood? What are the potential consequences of the blunted brain sparing response for neurodevelopment and cognitive function?

The discussion highlights the potential benefits and risks of melatonin treatment, it does not provide clear recommendations for clinical translation (PMID: 22325255). What would be the optimal timing, dosage, and duration of melatonin administration in human pregnancies complicated by FGR? How might the risks of rebound oxidative stress or transient endothelial dysfunction be mitigated in a clinical setting?

The discussion frequently speculates on potential mechanisms (e.g., rebound oxidative stress, delayed maturational pruning of receptors) without providing definitive evidence. While hypotheses are valuable, the lack of experimental validation weakens the conclusions.

END OF COMMENTS

Dear editors and reviewers,

We are grateful for the opportunity to address your comments and concerns, and sincerely appreciate the time dedicated to reviewing this article. Below I have responded to each comment, and we hope that the revisions are made aligned with the expectations of all editors and reviewers.

Sincerely,

Dr Charmaine Rock

Reviewing editor:

1. The supplementary data suggests that t tests and one way ANOVAs were used to analyse the data. It is not clear why a 2 way ANOVA was not used.

We appreciate this query on the statistics used, it informs us that we did not adequately explain why we selected our statistical approach. In this study, we had three variables – age, growth status and treatment; this would necessitate a 3-way ANOVA if all groups were filled. However, to reduce animal use we did not include a control group treated with melatonin (as this would not be clinically relevant), and therefore, we could not conduct a 3-way ANOVA. With statistical advice, we performed a two-step analysis, firstly assessing age changes between newborn and 4 weeks within each of the three groups (control, FGR, and FGR+MLT). We next carried out a one-way ANOVA within each age group to determine group differences, followed by a post hoc Tukey test to assess specific differences between groups. This design allowed us to address the study aims while accounting for the limitation of a lack of a control+MLT group. We have reworded the statistics section for greater clarity.

Line 262: In the first instance, we assessed gestational age changes separately for each parameter in the control, FGR and FGR+MLT groups using an unpaired T-test (or a Mann Whitney U test for data that was not normally distributed). This was followed by a one-way analysis of variance (ANOVA) with a post hoc Tukey test, where applicable on the same variables to determine the effects of the independent variables of treatment. When data was not normally distributed, a Kruskal Wallis test was used with a post hoc Dunn's test.

2. All data points should be shown as well as the mean and SD.

We have updated Figure 3 to include all requested data points. We understand and appreciate the importance of displaying all data points however, we believe that Figures 1A&B and Figures 2A&B are appropriate as this is standard expression of myography data and the journal's statistics policy states that: 'Note exceptions: 1) If each subject has numerous data points associated with it (e.g. time course data), we would treat 'n' as being each data point, not the number of subjects, and there can be flexibility in the format of presentation.' We believe this applies to Figures 1A&B and Figures 2A&B. Figures 1C&D and Figures 2C&D are derived from the nonlinear regression performed on the data from A&B of the corresponding figures. The output for this analysis only provides the mean and SEM or CI, therefore we have changed Figure 1C&D and Figure 2C&D to display mean and CI.

3. The methods do not include information about access to food/water, housing or pain relief.

Thank you for bringing this to our attention. This oversight has been rectified with the appropriate information added to the "Animal care and surgical preparation in the Methods.

Line 138: Singleton-bearing mixed breed pregnant ewes were delivered to Monash Health Translation Precinct Animal Facility and housed in individual pens in a room maintained at 22°C and 44-55% humidity with a 12-hour light/dark cycle. All ewes were fed ~1 kg lucerne chaff daily and had free access to water. Food intake and general wellbeing of the ewe were monitored daily, and any abnormal behaviour or symptoms were treated accordingly.

Line 151: *Additionally, all ewes received paracetamol for analgesia at the cessation of surgery (rectal, 500 mg) followed by repeated doses for 3 days (oral, 1g) (Panadol, Ireland).*

4. The postmortem describes collection of blood vessels but not the heart.

Thank you for bringing this to our attention. This information has been added to “Postmortem and tissue collection” in the Methods.

Line 174: *Whole hearts were fixed in 10% neutral-buffered formalin for assessment of cardiac dimensions.*

5. JP does not have supplementary data.

We have added one of the supplemental tables (S1) to the main manuscript and included the information from table S2 within the text of the manuscript.

6. The Discussion would benefit from consideration of previous literature in the field of FGR, blood pressure regulation and SNS from authors such as Ruitjenbeek, Danielson, Zhang, Darby, Gilbert, Philips (DIABETIC MEDICINE, 1997; 14: 673-677).

We have added additional discussion around blood pressure regulation and the SNS, and incorporated comments with respect to #7 below.

Line 475: *The sympathetic arm of the autonomic nervous system is critical for the maintenance of blood pressure and, importantly, in the context of FGR, mediating the brain sparing response via peripheral vasoconstriction.(Giussani, 2016) Previous preclinical studies have shown that FGR alters components of the sympathetic nervous system, including hyperinnervation of peripheral vascular beds of hypoxic chicken embryos(Ruijtenbeek et al., 2000) and increased plasma catecholamine concentration in growth-restricted fetal sheep.(Simonetta et al., 1997) Therefore, components of the sympathetic nervous system that are critical for mediate blood pressure regulation were interrogated to determine potential implications of melatonin administration. In the current study, all lambs were normotensive. This is in keeping with studies using the carunclectomy model of ovine FGR that have demonstrated that growth-restricted fetuses are normotensive at rest, but then respond with a greater decrease in blood pressure when α 1A-AR activity is blocked with phentolamine compared to control fetuses.(Danielson et al., 2005; Darby et al., 2021) In the current study, the abundance of α 1A-AR was not different between control and FGR lambs at either timepoint and both groups showed a pruning of α 1A-AR with age. In contrast, α 1A-AR abundance in the femoral artery was elevated in normotensive 4-week-old FGR+MLT lambs. Combined, these findings suggest that in FGR fetuses exposed to melatonin treatment, there is a greater reliance in α 1A-adrenergic activation for blood pressure regulation at 4-weeks of age. Activation of α 1A-AR occurs primarily with binding of catecholamines, which are released systemically by the adrenal medulla or locally by postganglionic neurons in the target artery to elicit vasoconstriction (Perez, 2021)*

7. Other studies of FGR in sheep have also shown no change in blood pressure.

We have incorporated information regarding previous ovine studies throughout the manuscript (e.g. response to query 6).

8. Line 555 refers to sleep patterns but these are not reported in the paper.

Given the chronic nature of melatonin exposure to the ewe and fetus, we believe that it is important to address sleep habits, albeit this was not a primary outcome of this study. We have added the wording “anecdotally, we did not observe...” to convey that we did not observe overt changes in the sleep states of the sheep in our studies. We have reworded this to improve clarity.

Line 862: Further, melatonin regulates circadian rhythm and is used as a sleep aid, and while anecdotally, we did not observe changes in sleep patterns in the melatonin-treated animals, we did not formally assess sleep patterns in the mum or lambs, and high dose melatonin to newborn lambs does increase their time spent sleeping.

Referee #1

- 1) I may have missed it, but I don't believe animal numbers were mentioned anywhere in the manuscript. In addition most of the graphs are simple bar graphs without individual data points making it difficult to gauge the power of the analyses.**

We apologise for this oversight. We had included the animal numbers in Table 1 (now Table 2), we have now also added these numbers to the figure legends for Figures 1 and 2. We have changed Figure 3 to show individual data points; however, this is not feasible for Figures 1 and 2, as discussed in response to the reviewing editor (comment 2).

- 2) Sex effects were mentioned in the discussion/limitation section, however, again this was not clarified in the methods the ratio of males to females within each group. This is particularly important, as many studies have reported sex-specific responses to melatonin treatment.**

The ratio of females to males is included in Table 1 (now Table 2) below the number of animals.

- 3) The Journal of Physiology policy is to use Standard deviation, not SEM unless properly justified.**

We have changed the SEM to SD throughout the manuscript to align with the journal's policy.

- 4) I don't think the weight gain measure at day1 provides any meaningful purpose in the study as outlined in the discussion section. It would be more meaningful if all animals were measured 24hrs after birth.**

We thank the reviewer for this comment, and we agree, given that the weights were not taken at 24 hours of age. As such we have removed any discussion of this outcome and edited the manuscript as below. All references to 24 hours after birth have been changed to the day after birth. Key examples are below.

Line 172 (methods): Lambs were weighed the day after birth.....

Line 276 (results): At postmortem, the day after birth, both newborn FGR and FGR+MLT.....

Minor comments:

- 5) Line 132: Antibiotics were mentioned in the methods, but no detail was provided on what was given.**

Thank you for noticing this error, we have included details on antibiotics in the methods.

Line 149: *Antibiotics (1 g Ampicillin: Austrapen, Lennon Healthcare, Australia; 5 mL Engemycin: Coopers, Australia) were administered to the ewe via the jugular vein immediately following the induction of anaesthesia and were maintained for three days post-surgery.*

- 6) Line 241: I believe this should say 'non-parametric' here when describing how data that was not normally distributed was analyzed.**

Thank you, this has been rectified.

- 7) Line 244: "FGR lambs less than control lambs" missing word? Weighed?**

Weighed is missing here, thank you for pointing this out.

8) Line 248: Missing P value "was significantly lower than that of controls in the same cohort."

This p-value has been added.

9) Line 255: again, missing P-value at the end of the sentence.

This p-value has been added.

10) Line 387: I think you mean symmetric not asymmetric here, as this is what happens in regular growth restriction.

Thank you, this has been rectified.

11) The thickness of the error bars in the "Maximal response to SNP" graph is smaller than the other graphs. Would it be possible to include individual data points on all graphs like in Figure 4? It helps with transparency. Bar graphs can be used to 'hide' variability in data.

We have changed the thickness of the error bars in this graph to match the others. Unfortunately, individual data points cannot be added to the bar graphs in Figures 1 & 2 as the data is derived from the nonlinear regression performed on the data presented in A&B and does not have individual data points (discussed further in response to the reviewing editor).

Referee #2

- 1. Despite consistently being described as a scavenger with antioxidant effects, direct evidence concerning this trait for melatonin is elusive. Consequently, its vascular effects potentially involve mainly receptor-mediated actions rather than direct targeting of oxidant agents (PMID: 22916799; 26514204; 21752095). This issue must be acknowledged in the introduction and discussion.**

Thank you for the valuable insight into our findings. We have amended mentions of antioxidant scavenging from the introduction and discussion to include information on this potential mechanism.

Line 91 (introduction): *Melatonin acts through receptor-mediated and receptor-independent pathways. Its receptor-mediated actions are driven by G-protein-coupled melatonin receptors (MT1 and MT2) in many target organs, which in blood vessels, are expressed in both the endothelium and smooth muscle (Liu et al., 2016). Activation of these receptors modulates redox homeostasis by upregulating enzymes such as superoxide dismutase and inhibiting pro-oxidant enzymes, including NADPH oxidase. Although melatonin is often described as a direct free radical scavenger, the in vivo evidence for this action is limited (Paulis et al., 2012; Liu et al., 2016; Monteiro et al., 2024), and current data suggest that its vascular protective actions are more plausibly explained by receptor-mediated enhancement of endogenous antioxidant defenses rather than direct interactions with reactive oxygen species.*

Line 619 (Discussion): *Although melatonin is frequently characterised as an antioxidant, direct evidence for its ROS-scavenging activity in vivo remains limited, and its vascular effects may be predominantly receptor-mediated. (Paulis et al., 2012; Liu et al., 2016; Monteiro et al., 2024) The chemical reactivity of superoxide with NO far outpaces its potential interaction with melatonin. (Huie & Padmaja, 1993; Zang et al., 1998) Instead, melatonin appears to exert a stronger effect by enhancing endogenous antioxidant systems, particularly through the upregulation of superoxide dismutase, which neutralises superoxide more efficiently than melatonin itself. (Morvaridzadeh et al., 2020) This distinction may help to explain our findings in FGR+MLT lambs, where melatonin treatment only partially reduced oxidative stress. The effect is consistent with reinforcement of antioxidant defences rather than direct actions of melatonin on elimination of reactive oxygen species.*

- 2. Myography results must be informed as most studies do. For instance, the maximal relaxing response is the difference between the active tone after and before adding the relaxing agent, but not the residual tone after stimulation. Please refer to more standardized protocols available if required (PMID: 16028678; 33989082).**

Thank you for this comment. We have changed the graphs for maximal relaxation to reflect the difference in active tone before and after addition of the relaxant, and made amendments to address this in the methods.

Line 189: *Concentration-response curves were graphed as percentage tension present in the vascular bed and analysed using an agonist-response line of best fit. The maximal relaxant response (R_{max}) was calculated from the line of best fit and expressed as the percentage difference between active tone before the addition of the vasodilator and after the administration of the highest dose of the vasodilator (10-5 mol/L of ACh or 10-4 mol/L SNP).*

- 3. Considering the low number of subjects studied, data in tables must be informed as the median and interquartile range, and statistics performed considering non-parametric test (PMID: 27556235). Similarly, individual values must be informed in figures 1 - 3, as shows figure 4.**

At the request of the reviewing editor, we have replaced SEM with SD throughout the manuscript which aligns with the statistics policy for the Journal of Physiology. In our analyses, we assessed the distribution of each dataset, and when normality was not met, non-parametric tests were applied. When normality assumptions were satisfied, we applied parametric tests, which is consistent with recent publications in this journal using sheep models with similar sample sizes (i.e. <https://doi.org/10.1113/JP288303> where n=6). We have added individual data points for Figure 3 but have provided a response to why this isn't possible for Figures 1&2 to the reviewing editor (comment #2).

- 4. The study provides detailed mechanistic insights, but some findings remain inconsistent or unexplained. For instance, melatonin improved NO response in newborn FGR lambs but did not improve overall vasodilatory capacity. The reasons for this are not fully addressed. Similarly, the transient endothelial dysfunction in newborn FGR+MLT lambs differs from previous studies showing improved vasodilation. Some potential reasons for these differences, such as dosage or timing of melatonin administration need to be explored.**

We thank the reviewer for highlighting these important points. We agree that the improvement in vasodilation in newborn FGR+MLT lambs is complex. We have amended the discussion around this finding to increase clarity. To address the reduction in vasodilatory capacity in newborn FGR+MLT lambs the following text was added.

Line 562: This reflects a shift in endothelial control of vascular tone, with greater reliance on NO-dependent mechanisms. Additionally, melatonin has been shown to inhibit voltage-gated potassium channels in peripheral arteries of rodents (Lew & Flanders, 1999), crucial for EDH-mediated responses, contributing to the reduction in EDH contribution following melatonin treatment. Consequently, overall endothelial vasodilation remains impaired because the enhancement of NO activity was insufficient to offset the loss of EDH-mediated vasodilation.

To address the differences between our vasodilatory outcomes with previous studies, we have added the following information to our discussion:

Line 527: However, there are also significant differences between these studies, including the dose of melatonin (6mg/day in the previous study vs 15mg/day in the current study), the vascular beds examined (which in the earlier study were likely not subjected to chronic in utero vasoconstriction), and the timing of treatment initiation (0.7 gestation previously versus 0.6 gestation in our study)(Tare et al., 2014) To the best of our knowledge, our study is the first to assess peripheral vascular function in newborns exposed to antenatal melatonin,(Rock et al., 2024) and the observation of perinatal endothelial dysfunction represents a novel finding. In the current study the dose of melatonin was selected to be pharmacologically comparable to the PROTECTMe clinical trial (Palmer et al., 2019) (30mg orally vs 15mg i.v.) supporting the clinical relevance of our exposure paradigm. The discrepancies in vascular outcomes between the current study and previous studies underscore the complexity of melatonin's effects on the cardiovascular system. Endothelial dysfunction during the perinatal period may be particularly

detrimental, as this is a time of cardiovascular adaptation, a risk heightened in growth-restricted neonates (Oyang et al., 2023).

- 5. While the study focuses on outcomes at 24 hours and 4 weeks of age, the discussion does not sufficiently address the long-term implications of the observed changes. Will the elevated oxidative stress and inflammation in 4-week-old FGR+MLT lambs persist into adulthood? What are the potential consequences of the blunted brain sparing response for neurodevelopment and cognitive function?**

We thank the reviewer for raising this important point. Our study was designed to assess outcomes at 24 hours and 4 weeks of age, and therefore, we cannot make definitive claims about the persistence of oxidative stress, inflammation, or vascular changes into later life, although we are happy to speculate. Previous studies in both preclinical models and human cohorts of FGR suggest that early-life oxidative stress, inflammation, and altered cerebrovascular adaptation may contribute to long-term cardiovascular and neurodevelopmental vulnerability. We have now expanded the discussion to note that while our findings highlight early mechanisms of risk, future studies with longer follow-up are needed to determine whether these changes persist into adulthood and whether the blunted brain-sparing response has lasting cognitive or developmental consequences.

Line 585: Inflammation is widely recognised as a precursor to cardiovascular disease, with elevated levels of CRP associated with an increased risk of cardiovascular events. (Sakkinen et al., 2002) In our study, antenatal melatonin reduced indices of oxidative stress, yet was associated with persistently high vascular CRP at 4 weeks after birth. This paradoxical finding is novel and warrants further longer-term follow-up, given that elevated CRP is a biomarker of vascular inflammation and a predictor of adult-onset atherosclerosis and cardiovascular disease. (Frery et al., 2020) In rodent models, chronic melatonin supplementation has been reported to improve long-term vascular reactivity (Hansell et al., 2022), but whether these benefits are also seen in larger-animal models and humans or persist when treatment ceases at birth remains unknown. Accordingly, the implications of sustained vascular inflammation in FGR+MLT lambs remain unresolved and requires further study into adolescence and adulthood.

In these same lambs, neuropathology and neurocognitive outcomes have also been assessed; however, these data will be published separately. Our unpublished results indicate that antenatal melatonin consistently improves neuropathology and cognitive function throughout the first 4 weeks of life, but these findings are beyond the scope of this manuscript. We have added the following sentence to address this comment:

Line 469: The implications of modifying or abolishing the brain sparing response through peripheral vasodilation are not known, but are certainly of interest. Melatonin is shown to have protective benefits for the FGR brain in fetal and newborn sheep, by increasing myelination and preventing axonal damage, (Miller et al., 2014; Malhotra et al., 2024) and, whether this improves the trajectory of brain development in FGR (Dudink et al., 2025) needs further consideration.

- 6. The discussion highlights the potential benefits and risks of melatonin treatment, it does not provide clear recommendations for clinical translation (PMID: 22325255). What would be the optimal timing, dosage, and duration of melatonin administration in human pregnancies**

complicated by FGR? How might the risks of rebound oxidative stress or transient endothelial dysfunction be mitigated in a clinical setting?

We agree that clinical translation of melatonin treatment is important; however, much of the existing research has focused on its neuroprotective effects, and our understanding of its impact on the cardiovascular system remains insufficient to indicate the use of melatonin specifically as a cardiovascular protectant – indeed our results further indicate the complexities of mediating fetal/antenatal cardiovascular development. Further studies will be necessary to clarify the longer-term effects of antenatal melatonin on cardiovascular wellbeing. Our findings suggest potential cardiovascular benefits but also highlight the possibility of off-target effects such as rebound oxidative stress and persistent vascular inflammation. Importantly, these data indicate that ongoing clinical studies such as the ProtectMe trial, which is primarily designed to evaluate the neuroprotective benefits of melatonin, should also incorporate short- and long-term postnatal cardiovascular outcomes to ensure a comprehensive assessment of both efficacy and safety.

Line 596: Melatonin is a well-established neuroprotective agent in perinatal compromise, including for FGR, with melatonin currently under investigation in human pregnancies complicated by FGR in the PROTECTMe trial (Palmer et al., 2019). Our findings highlight the importance of extending such assessments beyond the brain to include postnatal cardiovascular endpoints. FGR affects multiple organ systems, with pathology evident in the cardiovascular system. While our findings support potential cardiovascular benefits, they also highlight the need for careful evaluation before clinical translation can be recommended.

- 7. The discussion frequently speculates on potential mechanisms (e.g., rebound oxidative stress, delayed maturational pruning of receptors) without providing definitive evidence. While hypotheses are valuable, the lack of experimental validation weakens the conclusions.**

We appreciate this comment; it is very valid. This study has revealed several intriguing results, which have in turn raised additional questions about the biological actions of melatonin on cardiovascular development. We acknowledge that some of the proposed mechanisms remain speculative as they were not directly tested in this study. We have altered the discussion to reflect this point and to guide future research aimed at validating these mechanisms.

Line 627: This study is among the first to demonstrate the postnatal cardiovascular implications of antenatal melatonin administration, with assessments conducted at multiple postnatal timepoints. Our findings in the femoral artery raise important questions around the potential for rebound inflammatory or oxidative stress following the withdrawal of exogenous melatonin at birth. Further work is needed to define the pathways that drive these changes in inflammation and oxidative stress, and whether vascular inflammation at 4-weeks persists to adulthood, together with the cardiovascular implications.

Dear Dr Rock,

Re: JP-RP-2025-288750R1 "Antenatal melatonin for cardiovascular deficits in fetal growth restriction" by Charmaine R Rock, Tegan A White, Beth R Piscopo, Amy E Sutherland, Yen Pham, Emily J Camm, Connor Karozis, Fiona L Cousins, Suzanne L Miller, and Beth J Allison

Thank you for submitting your manuscript to The Journal of Physiology. It has been assessed by a Reviewing Editor and by 2 expert referees and we are pleased to tell you that it is acceptable for publication following satisfactory revision.

REVISION CHECKLIST:

Please upload two versions of your manuscript text: one with all relevant changes highlighted and one clean version with no changes tracked. The manuscript file should include all tables and figure legends, but each figure/graph should be uploaded as separate, high-resolution files. The journal is now integrated with Wiley's Image Checking service. For further details, see: <https://www.wiley.com/en-us/network/publishing/research-publishing/trending-stories/upholding-image-integrity-wileys->

image-screening-service

We look forward to receiving your revised submission.

Yours sincerely,

Laura Bennet
Senior Editor
The Journal of Physiology

EDITOR COMMENTS

Reviewing Editor:

Thank you for revising the paper.

Please include sample size in the methods. This should be included at the first description of the groups. If analysis of any measure is not performed in all animals, please indicate the sample size used. Sample size should also be indicated in figure legends and tables.

REFEREE COMMENTS

Referee #1:

Thank you for addressing my comments. I have no further suggestions.

Referee #2:

All the comments have been addressed by the authors

END OF COMMENTS

Dear editors and reviewers,

Thank you for your valuable time and insightful feedback, which have significantly enhanced the quality of this manuscript. In response to the final comment from the reviewing editor, we have included detailed sample size information in the initial description of the groups in the Methods section, as well as in Table 3 and the legend for Figure 4.

Sincerely,

Dr. Charmaine Rock

Dear Dr Rock,

Re: JP-RP-2025-288750R2 "Antenatal melatonin for cardiovascular deficits in fetal growth restriction" by Charmaine R Rock, Tegan A White, Beth R Piscopo, Amy E Sutherland, Yen Pham, Emily J Camm, Connor Karozis, Fiona L Cousins, Suzanne L Miller, and Beth J Allison

We are pleased to tell you that your paper has been accepted for publication in The Journal of Physiology.

Yours sincerely,

Laura Bennet
Senior Editor
The Journal of Physiology

IMPORTANT POINTS TO NOTE FOLLOWING ACCEPTANCE OF YOUR PAPER:

- You can help your research get the attention it deserves! Check out Wiley's free Promotion Guide for best-practice recommendations for promoting your work at: www.wileyauthors.com/eoo/guide. You can learn more about Wiley Editing Services which offers professional video, design, and writing services to create shareable video abstracts, infographics, conference posters, lay summaries, and research news stories for your research at: www.wileyauthors.com/eoo/promotion.

- If you would like to receive our 'Research Roundup', a monthly newsletter highlighting the cutting-edge research published in The Physiological Society's family of journals (The Journal of Physiology, Experimental Physiology, Physiological Reports, The Journal of Nutritional Physiology and The Journal of Precision Medicine: Health and Disease), please click this link, fill in your name and email address and select 'Research Roundup': <https://www.physoc.org/journals-and-media/membernews>

EDITOR COMMENTS

Reviewing Editor:

Thank you for revising the paper.